# Impute-MACFM: Imputation based on Mask-Aware Flow Matching

## Abstract

Tabular data are central to many applications, especially longitudinal data in healthcare, where missing values are common, undermining model fidelity and reliability. Prior imputation methods either impose restrictive assumptions or struggle with complex cross-feature structure, while recent generative approaches suffer from instability and costly inference. We propose Impute-MACFM, a mask-aware conditional flow matching framework for tabular imputation that is robust under standard missingness mechanisms: missing completely at random (MCAR), missing at random (MAR), and missing not at random (MNAR). Its mask-aware objective builds trajectories only on missing entries while constraining predicted velocity to remain near zero on observed entries, using flexible nonlinear schedules. Impute-MACFM combines: (i) stability penalties on observed positions, (ii) consistency regularization enforcing local invariance, and (iii) time-decayed noise injection for numeric features. Inference uses constraint-preserving ordinary differential equation integration with per-step projection to fix observed values, optionally aggregating multiple trajectories for robustness. Across diverse benchmarks, Impute-MACFM achieves state-of-the-art or competitive results while delivering more robust, efficient, and higher-quality imputation than competing approaches, supporting flow matching as a promising direction for tabular missing-data problems, including longitudinal data.

## 1 Introduction

Tabular data with missing values is ubiquitous across critical domains from healthcare records where patient measurements are irregularly collected, to financial systems with incomplete transaction histories, to scientific experiments with sensor failures. In medical settings alone, electronic health records routinely exhibit 20–80% missingness (Wells et al., 2013), severely limiting the applicability of machine learning models that require complete data. While the importance of accurate imputation is well-recognized, existing methods face a fundamental trade-off between computational efficiency and imputation quality, particularly when dealing with the heterogeneous nature of real-world tabular data that mixes continuous, categorical, and ordinal features.

Traditional statistical approaches such as mean imputation and MICE (van Buuren & Groothuis-Oudshoorn, 2011) provide computationally efficient solutions but rely on strong distributional assumptions that rarely hold in practice. These methods fail to capture complex non-linear dependencies between features, leading to biased downstream analyses. Machine learning methods like Miss-Forest (Stekhoven & Bühlmann, 2012) improve upon statistical approaches by modeling non-linear relationships through ensemble learning, yet they provide only point estimates without uncertainty quantification critical for high-stakes applications.

The deep learning revolution brought more expressive models: GANs (GAIN (Yoon et al., 2018)) generate realistic imputations through adversarial training, VAEs (MIWAE (Mattei & Frellsen, 2019)) provide probabilistic frameworks with uncertainty estimates, and Transformers (TabTransformer (Huang et al., 2020), SAINT (Somepalli et al., 2021)) capture long-range feature dependencies through attention mechanisms. However, these methods suffer from training instability (GANs), posterior collapse (VAEs), or excessive data requirements (Transformers), limiting their practical deployment.

Recently, diffusion models have emerged as the state-of-the-art for generative modeling, with methods like CSDI (Tashiro et al., 2021) and TabDDPM (Kotelnikov et al., 2023) demonstrating superior imputation quality through iterative denoising and outperforming classical baselines (e.g., mean, MICE, and matrix factorization) on challenging missingness patterns. Yet diffusion models introduce a new computational bottleneck: they require hundreds to thousands of neural network evaluations for each imputation, making them prohibitively expensive for large-scale applications. Moreover, their Gaussian noise assumption poorly matches categorical features common in tabular data, and maintaining observed values as hard constraints throughout the reverse process requires careful engineering that often breaks theoretical guarantees.

Flow matching (Lipman et al., 2023; Albergo et al., 2023) offers a promising alternative by directly learning deterministic transformations between distributions through ordinary differential equations (ODEs). Unlike diffusion's stochastic dynamics, flow matching provides: (1) simulation-free training with closed-form target vector fields, (2) flexible interpolation paths not restricted to diffusion processes, and (3) efficient inference requiring 10–50 ODE steps versus 100–1000 for diffusion. However, naively applying flow matching to tabular imputation fails to address several critical challenges: generic formulations do not explicitly enforce consistency between the interpolation schedule and the target velocity field, standard interpolation paths ignore the heterogeneous nature of tabular features, and the framework lacks mechanisms to preserve observed values as hard constraints.

In this work, we introduce Impute-MACFM, a mask-aware conditional flow matching framework specifically designed for efficient and accurate tabular imputation. Our work bridges the gap between modern generative modeling and practical tabular imputation requirements. We derive a mask-aware, schedule-consistent conditional flow matching formulation that: (1) defines a velocity field that properly accounts for non-linear schedule functions in the interpolation path and ensures consistency between the schedule parameterization and the learned vector field on the target coordinates; (2) designs specialized interpolation paths that only transport missing entries while respecting the heterogeneous nature of tabular data; (3) ensures hard constraint satisfaction through a simple projection-based ODE integration scheme; and (4) achieves 5–10× faster inference than diffusion methods while maintaining or improving imputation quality. Throughout, we follow standard MCAR/MAR/MNAR masking schemes used in recent tabular benchmarks and show that Impute-MACFM remains robust under these mechanisms, rather than attempting to explicitly model the missingness process itself. By combining the advantages of flow matching with careful architectural design for tabular data, our method provides a principled and efficient solution to missing value imputation that scales to real-world applications.

## 2 RELATED WORK

**Autoencoder-based methods** learn compressed representations for imputation. DAE (Vincent et al., 2008) reconstructs complete data from corrupted inputs, while variational variants like MIWAE (Mattei & Frellsen, 2019) and notMIWAE (Ipsen et al., 2021) provide importance-weighted bounds for principled uncertainty estimation. HI-VAE (Nazabal et al., 2020) specifically handles heterogeneous tabular data through specialized encoder-decoder architectures. While these methods offer theoretical guarantees, they often struggle with complex missing patterns and may suffer from posterior collapse.

**GAN-based approaches** leverage adversarial training for realistic imputation. GAIN (Yoon et al., 2018) introduces a discriminator to distinguish observed from imputed values, while PC-GAIN (Wang et al., 2021) incorporates causal discovery. MisGAN (Li et al., 2019) adds a missingness generator to model the mechanism explicitly. Despite generating sharp imputations, GANs suffer from mode collapse and training instability, limiting their reliability.

**Transformer architectures** have shown promise for tabular data by treating features as tokens. Beyond standard adaptations like TabTransformer (Huang et al., 2020) and SAINT (Somepalli et al., 2021), recent work includes TabPFN (Hollmann et al., 2023) for few-shot learning and TransTab (Wang & Sun, 2022) for transfer learning across datasets. TDI (He et al., 2023b) specifically targets distributional imputation through specialized attention mechanisms. While powerful, these methods require substantial training data and struggle with numerical features that lack natural token representations.

**Diffusion models for missing data.** Diffusion models have emerged as state-of-the-art for imputation tasks. CSDI (Tashiro et al., 2021) pioneers conditional diffusion for time series, learning to denoise while conditioning on observed values. TabDDPM (Kotelnikov et al., 2023) adapts multinomial diffusion for categorical features in tabular data. MissDiff (Yidong et al., 2023) further tailors score-based diffusion to tabular data with missing entries, combining masking strategies with diffusion dynamics. Recent innovations such as DiffPuter (Hengrui et al., 2025) integrate diffusion-style iterative refinement with likelihood-based updates. The primary limitation remains computational cost, diffusion models typically require hundreds of denoising steps, making them impractical for large-scale or real-time applications, and their stochastic nature complicates enforcing hard constraints on observed values.

**Flow-Based Generative Models**. Normalizing flows provide exact likelihood computation through invertible transformations. Early work focused on architectural innovations: RealNVP (Dinh et al., 2016), Glow (Kingma & Dhariwal, 2018), and Neural Spline Flows (Durkan et al., 2019) increased expressiveness while maintaining invertibility. However, these methods require specialized architectures that limit flexibility.

Flow matching (Lipman et al., 2023; Albergo et al., 2023) relaxes the invertibility constraint, directly learning ODE vector fields through regression. This enables arbitrary architectures and more efficient training. Rectified Flow (Liu et al., 2023) demonstrates that straight paths minimize transport cost, while conditional variants (Tong et al., 2024) extend to conditional generation. Riemannian Flow Matching (Chen & Lipman, 2024) generalizes to manifold-valued data.

To our knowledge, we provide the first systematic study of applying mask-aware flow matching to tabular imputation, introducing domain-specific designs for heterogeneous, incomplete, and longitudinal data.

## 3 METHOD

We propose a mask-aware conditional flow matching (Impute-MACFM) framework shown in figure 1 for tabular imputation. The key idea is to separate features into observed, conditioning, and target partitions and to drive only the target subset along a schedule-controlled path from noise to data while strictly preserving observed entries. A schedule-consistent velocity field provides the correct training target under linear or non-linear schedules, and two lightweight regularizers stabilize conditioning dimensions and encourage local smoothness. At inference, a constraint preserving ODE solver (Euler/Heun with projection) generates imputations that respect all observed constraints.

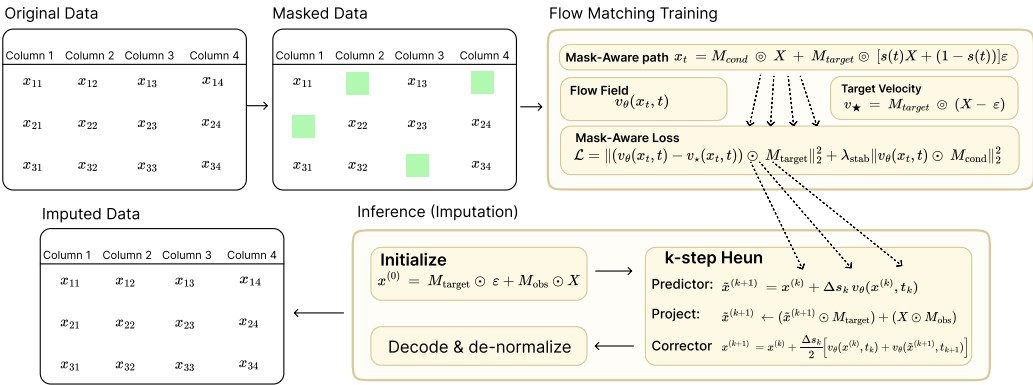

Figure 1: Flow Matching and Imputation Process

### 3.1 IMPUTE-MACFM: MASK-AWARE CONDITIONAL FLOW MATCHING

Unlike standard flow matching (Lipman et al., 2023; Albergo et al., 2023) that applies a uniform path across all dimensions, our formulation of our Impute-MACFM framework for heterogeneous tabular data differentiates conditioning and target regions to respect their semantics.

Our method considers single–time–point tabular imputation where each sample $\mathbf{X} \in \mathbb{R}^D$ may contain missing entries. Following conditional flow matching (Lipman et al., 2023) and DiffPuter (Hengrui et al., 2025), we partition the feature space using three binary masks:

- $M_{\text{obs}} \in \{0,1\}^D$: observed entries not used for conditioning or supervision
- $M_{\text{cond}} \in \{0,1\}^D$: observed entries used as conditioning context
- $M_{\text{tgt}} \in \{0,1\}^D$: entries to be imputed (held out during training; truly missing at inference)

These sets are pairwise disjoint with $M_{\text{obs}} + M_{\text{cond}} + M_{\text{tgt}} = \mathbf{1}$. During training, $M_{\text{cond}}$ and $M_{\text{tgt}}$ are sampled within observed locations to provide supervised targets; during inference, $M_{\text{obs}}$ corresponds to actually observed entries and $M_{\text{tgt}}$ to truly missing entries. Our goal is to model $p\big(\mathbf{X}_{M_{\text{tgt}}} \mid \mathbf{X}_{M_{\text{obs}} \cup M_{\text{cond}}}\big)$ while strictly preserving observed constraints.

**Conditional Path**. Given a monotonic schedule $s : [0,1] \to [0,1]$ with $s(0) = 0$, $s(1) = 1$, and $s'(t) > 0$ for all $t \in (0,1)$, we define the conditional path:

$$\mathbf{x}_t = M_{\text{cond}} \odot \mathbf{X} + M_{\text{tgt}} \odot \big[s(t)\mathbf{X} + (1 - s(t))\boldsymbol{\varepsilon}\big] \tag{1}$$

where $\boldsymbol{\varepsilon} \sim \mathcal{N}(0, \mathbf{I})$, $t \in [0,1]$, and $\odot$ denotes the Hadamard product. This construction ensures three properties:

1. *Conditional preservation*: dimensions in $M_{\text{cond}}$ remain anchored to their observed values
2. *Smooth interpolation*: target dimensions follow a continuous path from noise to data
3. *Constraint satisfaction*: observed dimensions are enforced via projection at each solver step

We consider three schedules for interpolation dynamics: Linear $s(t) = t$ with $s'(t) = 1$; Power $s(t) = t^\gamma$ with $\gamma \in [1,3]$ and $s'(t) = \gamma t^{\gamma-1}$; and Cosine $s(t) = \frac{1}{2}(1 - \cos(\pi t))$ with $s'(t) = \frac{\pi}{2}\sin(\pi t)$. Non-linear schedules allocate more computation to later stages where signal-to-noise ratio is higher, empirically improving generation quality (Song et al., 2021b).

**Impute-MACFM: Schedule-Consistent Velocity Field**  When using non-linear schedules $s(t)$, the target velocity must account for the schedule derivative. Differentiating Equation 1 with respect to time gives the correct target velocity in $t$-space:

$$\mathbf{v}_\star(\mathbf{x}_t, t) = \frac{\partial \mathbf{x}_t}{\partial t} = M_{\text{tgt}} \odot s'(t)\left(\mathbf{X} - \boldsymbol{\varepsilon}\right). \tag{2}$$

This schedule-aware target is what we use in training and is crucial for stability under non-linear $s(t)$.

## 3.2 IMPUTE-MACFM: TRAINING OBJECTIVE

We train a neural velocity field $\mathbf{v}_\theta$ with a mask-aware flow matching loss on target dimensions (normalized by the number of target entries):

$$\mathcal{L}_{\text{FM}}(\theta) = \mathbb{E}_{\mathbf{X}, \boldsymbol{\varepsilon}, t}\left[\frac{\left\|\left(\mathbf{v}_\theta(\mathbf{x}_t, t) - \mathbf{v}_\star^{(t)}(\mathbf{x}_t, t)\right) \odot M_{\text{tgt}}\right\|_2^2}{\sum M_{\text{tgt}} + \epsilon}\right], \tag{3}$$

where $t \sim \mathcal{U}[0,1]$ (or Beta) and $\mathbf{v}_\star^{(t)}$ follows Equation 2.

To prevent drift on conditional dimensions we add a stability regularizer on $M_{\text{cond}}$:

$$\mathcal{R}_{\text{stab}}(\theta) = \mathbb{E}_{\mathbf{X}, \boldsymbol{\varepsilon}, t}\left[\|\mathbf{v}_\theta(\mathbf{x}_t, t) \odot M_{\text{cond}}\|_2^2\right]. \tag{4}$$

We further encourage local smoothness via a z-score regularizer on target dimensions, where $\boldsymbol{\xi} \sim \mathcal{N}(0, \mathbf{I})$ and the perturbation magnitude decays with $(1 - s(t))$:

$$\mathcal{R}_{\text{cons}}(\theta) = \mathbb{E}\left[\left\|\left(\mathbf{v}_\theta(\mathbf{x}_t + \eta_{\text{cons}}(1 - s(t))\boldsymbol{\xi}, t) - \mathbf{v}_\theta(\mathbf{x}_t, t)\right) \odot M_{\text{tgt}}\right\|_2^2\right]. \tag{5}$$

---

**Algorithm 1** Mask-Aware Flow Matching Training

---

**Require:** Dataset $\mathcal{D}$ of $(\mathbf{X}, M_{\text{obs}}, M_{\text{cond}}, M_{\text{tgt}})$; schedule $s$; hyperparams $\lambda_{\text{stab}}, \lambda_{\text{cons}}, \eta_{\text{cons}}, \sigma_{\text{in}}$
1: **for** minibatch $(\mathbf{X}, M_{\text{obs}}, M_{\text{cond}}, M_{\text{tgt}})$ **do**
2:     Sample $t \sim \mathcal{U}[0, 1]$ (or Beta); compute $s(t), s'(t)$
3:     If $\sigma_{\text{in}} > 0$, add numeric-only input noise: $\mathbf{X} \leftarrow \mathbf{X} + \sigma_{\text{in}} (1 - s(t)) \, \boldsymbol{\xi}$ on $M_{\text{obs}}$
4:     Sample $\boldsymbol{\varepsilon} \sim \mathcal{N}(0, \mathbf{I})$; set $\mathbf{x}_t = M_{\text{cond}} \odot \mathbf{X} + M_{\text{tgt}} \odot \left( s(t)\mathbf{X} + (1 - s(t))\boldsymbol{\varepsilon} \right)$
5:     Set $\mathbf{v}_\star = M_{\text{tgt}} \odot s'(t) \, (\mathbf{X} - \boldsymbol{\varepsilon})$; get $\mathbf{v}_\theta = \mathbf{v}_\theta(\mathbf{x}_t, t)$
6:     $\mathcal{L}_{\text{FM}} = \left\| \left( \mathbf{v}_\theta - \mathbf{v}_\star \right) \odot M_{\text{tgt}} \right\|_2^2 / \left( \sum M_{\text{tgt}} + \epsilon \right)$
7:     $\mathcal{R}_{\text{stab}} = \left\| \mathbf{v}_\theta \odot M_{\text{cond}} \right\|_2^2 / \left( \sum M_{\text{cond}} + \epsilon \right)$
8:     If $\lambda_{\text{cons}} > 0$: set $\mathbf{x}_t' = \mathbf{x}_t + \eta_{\text{cons}} (1 - s(t)) \, \boldsymbol{\xi}$; $\mathbf{v}_\theta' = \mathbf{v}_\theta(\mathbf{x}_t', t)$; $\mathcal{R}_{\text{cons}} = \left\| \left( \mathbf{v}_\theta' - \mathbf{v}_\theta \right) \odot M_{\text{tgt}} \right\|_2^2 / \left( \sum M_{\text{tgt}} + \epsilon \right)$
9:     Minimize $\mathcal{L}_{\text{FM}} + \lambda_{\text{stab}} \mathcal{R}_{\text{stab}} + \lambda_{\text{cons}} \mathcal{R}_{\text{cons}}$
10: **end for**

---

The overall objective is

$$\mathcal{L}_{\text{total}}(\theta) = \mathcal{L}_{\text{FM}}(\theta) + \lambda_{\text{stab}} \, \mathcal{R}_{\text{stab}}(\theta) + \lambda_{\text{cons}} \, \mathcal{R}_{\text{cons}}(\theta), \tag{6}$$

where $\lambda_{\text{stab}}, \lambda_{\text{cons}} > 0$ are tuned on validation data. In addition, we apply a lightweight input augmentation only on numeric features: $\mathbf{X} \leftarrow \mathbf{X} + \sigma_{\text{in}} (1 - s(t)) \, \boldsymbol{\xi}$ at observed positions, which improves robustness without violating conditional semantics.

### 3.3 IMPUTE-MACFM: CONSTRAINT-PRESERVING INFERENCE

During inference, we solve the learned ODE in time space using a modified Heun predictor-corrector method with constraint projection. We use a uniform time grid $\{t_k\}_{k=0}^K$ with $t_k = k/K$ and $\Delta t = 1/K$.

*Initialization:* $\mathbf{x}^{(0)} = M_{\text{tgt}} \odot \boldsymbol{\varepsilon} + M_{\text{obs}} \odot \mathbf{X}$

For each step $k = 0, \ldots, K - 1$:

$$\text{Predictor:} \quad \tilde{\mathbf{x}}^{(k+1)} = \mathbf{x}^{(k)} + \Delta t \cdot \mathbf{v}_\theta(\mathbf{x}^{(k)}, t_k) \tag{7}$$

$$\text{Project:} \quad \tilde{\mathbf{x}}^{(k+1)} \leftarrow \tilde{\mathbf{x}}^{(k+1)} \odot M_{\text{tgt}} + \mathbf{X} \odot M_{\text{obs}} \tag{8}$$

$$\text{Corrector:} \quad \mathbf{x}^{(k+1)} = \mathbf{x}^{(k)} + \frac{\Delta t}{2} [\mathbf{v}_\theta(\mathbf{x}^{(k)}, t_k) + \mathbf{v}_\theta(\tilde{\mathbf{x}}^{(k+1)}, t_{k+1})] \tag{9}$$

$$\text{Project:} \quad \mathbf{x}^{(k+1)} \leftarrow \mathbf{x}^{(k+1)} \odot M_{\text{tgt}} + \mathbf{X} \odot M_{\text{obs}} \tag{10}$$

The dual projection ensures observed constraints are maintained while Heun's method reduces truncation error compared to Euler integration (Song et al., 2021a).

For non-uniform schedules, we construct an adaptive partition: $s_k = s(k/K)$, concentrating steps near $s = 1$ where vector field curvature is typically higher.

### 3.4 IMPUTE-MACFM: BACKBONE ARCHITECTURE

We primarily use a compact Multilayer Perceptron (MLP) backbone tailored for tabular flow matching. The input features $\mathbf{x} \in \mathbb{R}^D$ are projected to $\mathbb{R}^{d_{\text{model}}}$, and a sinusoidal time encoding is mapped to the same dimension and added additively.

**Impute-MACFM: MLP Backbone**    Time conditioning uses sinusoidal positional encoding followed by a small MLP:

$$\mathbf{c}_t = \text{MLP}(\text{SinusoidalPE}(t)) \tag{11}$$

The backbone then applies several SiLU MLP blocks (no dropout) to $\text{Linear}(\mathbf{x}) + \mathbf{c}_t$. The velocity head is a simple two-layer MLP mapping $\mathbb{R}^{d_{\text{model}}}$ to $\mathbb{R}^D$.

---

**Algorithm 2** Constraint-Preserving ODE Inference

---

**Require:** $\mathbf{X}_{\text{obs}}, M_{\text{obs}}, M_{\text{cond}}, M_{\text{tgt}}$; steps $K$; solver $\in \{\text{EULER}, \text{HEUN}\}$

1: $M_{\text{keep}} \leftarrow M_{\text{obs}} + M_{\text{cond}}$                           $\triangleright$ kept (disjoint) mask
2: Initialize $\mathbf{x}^{(0)} \leftarrow M_{\text{tgt}} \odot \boldsymbol{\varepsilon} + M_{\text{keep}} \odot \mathbf{X}_{\text{obs}}$
3: **for** $k = 0$ **to** $K - 1$ **do**
4:      $t_k \leftarrow k/K; \quad \Delta t \leftarrow 1/K$
5:      **if** solver $=$ EULER **then**
6:          $\mathbf{v} \leftarrow \mathbf{v}_\theta(\mathbf{x}^{(k)}, t_k)$
7:          $\tilde{\mathbf{x}} \leftarrow \mathbf{x}^{(k)} + \Delta t\, \mathbf{v}$
8:      **else**                                  $\triangleright$ Heun (improved Euler)
9:          $\mathbf{v}_1 \leftarrow \mathbf{v}_\theta(\mathbf{x}^{(k)}, t_k)$
10:          $\hat{\mathbf{x}} \leftarrow \mathbf{x}^{(k)} + \Delta t\, \mathbf{v}_1$                   $\triangleright$ predictor
11:          $\hat{\mathbf{x}} \leftarrow \hat{\mathbf{x}} \odot (1 - M_{\text{keep}}) + \mathbf{X}_{\text{obs}} \odot M_{\text{keep}}$      $\triangleright$ projection
12:          $\mathbf{v}_2 \leftarrow \mathbf{v}_\theta(\hat{\mathbf{x}}, t_{k+1})$
13:          $\tilde{\mathbf{x}} \leftarrow \mathbf{x}^{(k)} + \frac{\Delta t}{2}(\mathbf{v}_1 + \mathbf{v}_2)$             $\triangleright$ corrector
14:      **end if**
15:      $\mathbf{x}^{(k+1)} \leftarrow \tilde{\mathbf{x}} \odot (1 - M_{\text{keep}}) + \mathbf{X}_{\text{obs}} \odot M_{\text{keep}}$     $\triangleright$ hard-keep observed & conditioning
16: **end for**
17: **return** $\mathbf{x}^{(K)}$

---

**Optional Transformer Backbone.** As an alternative, we also support a Transformer variant (Vaswani et al., 2017) that treats each feature as a token and conditions with AdaLN-Zero (Peebles & Xie, 2023), using SwiGLU feed-forward layers (Shazeer, 2020). The per-token velocity head is zero-initialized to start from near-zero velocity. We found both backbones to be interchangeable in our pipeline; we default to the MLP for simplicity and efficiency.

# 4 IMPUTE-MACFM: EXPERIMENTS

This section describes the datasets, experimental protocol, baselines, implementation, and our evaluation and reporting choices. Full dataset statistics and preprocessing details are deferred to Appendix B.

## 4.1 DATASETS

We evaluate on eight public tabular datasets that span binary classification, multiclass classification, and regression: *Adult*, *Default of Credit Card Clients*, *MAGIC Gamma Telescope*, *Online Shoppers Purchasing Intention*, *Online News Popularity*, *Gesture Phase Segmentation*, *Letter Recognition*, and *Dry Bean*. We also include three private NIH datasets, *AHEI_2010_S1*, *AHEI_2010_S3*, and *AHEI_2010_S4*. Links and preprocessing for the public datasets as well as the private cohort descriptions are provided in Appendix B.

## 4.2 EXPERIMENTAL PROTOCOL

We use a standardized setup for all datasets. Each dataset is split into 70% training and 30% test with a fixed random seed; the same split is used for all methods. Models are fitted on the training split, and we report metrics on the training split ("in-sample") and on the held-out test split ("out-of-sample"), following common practice in prior work (e.g., Hengrui et al., 2025). We synthesize missingness at nominal rates of 30%, 50%, and 70%. For every mechanism–rate pair we draw ten random masks and report the mean and standard deviation across masks. Imputation quality on numeric features is measured by MAE and RMSE in both in-sample and out-of-sample settings. We evaluate Impute-MACFM under three standard missingness mechanisms: MCAR, MAR, and MNAR (Little & Rubin, 2019; Schafer, 1997; 1999; Schafer & Graham, 2002; Fang, 2017). In all cases we follow the generators used in Transformed Distribution Matching (TDM) (He et al., 2023a) and DiffPuter (Hengrui et al., 2025), reusing their public implementations. Our goal is to assess robustness under these established schemes, not to explicitly model the missingness process itself.

**MCAR.** For MCAR, each entry is dropped independently with probability $\rho$, i.e., $\Pr(R_{ij} = 0 \mid \mathbf{X}) = \rho$, $\Pr(R_{ij} = 1 \mid \mathbf{X}) = 1 - \rho$, with $\rho \in \{0.3, 0.5, 0.7\}$ in our experiments, exactly as in He et al. (2023a); Hengrui et al. (2025).

**MAR.** For MAR, missingness depends only on always-observed features. We split indices into $A$ (always observed) and $B$ (potentially missing) as in He et al. (2023a); Hengrui et al. (2025), and for $j \in B$ use $\Pr(R_{ij} = 0 \mid \mathbf{X}_i) = \sigma(\alpha_j + \boldsymbol{\beta}_j^\top \mathbf{X}_i^A)$, with $\alpha_j$ tuned to reach the target missing rate and $R_{ij} = 1$ for all $j \in A$.

**MNAR.** For MNAR, missingness depends on partially observed covariates. We adopt the first MNAR generator of He et al. (2023a): split indices into $C$ and $D$, draw an auxiliary MCAR mask $Z$ on $C$ with rate $\rho$, form $\tilde{\mathbf{X}}_i^C = Z_i^C \odot \mathbf{X}_i^C$, and for $j \in D$ define $\Pr(R_{ij} = 0 \mid \mathbf{X}_i) = \sigma(\alpha_j + \boldsymbol{\beta}_j^\top \tilde{\mathbf{X}}_i^C)$, again tuning $\alpha_j$ to match the target missing rate. Since $\tilde{\mathbf{X}}_i^C$ is itself masked, this falls into the MNAR setting of TDM and DiffPuter.

**Training and evaluation.** For all three mechanisms, train and test masks are generated with the same public code base as He et al. (2023a); Hengrui et al. (2025). These masks determine which entries are removed before imputation; all methods, including baselines, are evaluated under the same masks, ensuring a fair MCAR/MAR/MNAR comparison.

## 4.3 BASELINES

We compare against a broad set of imputation methods spanning statistical/optimization, classical ML, optimal-transport, and deep generative/diffusion models.

Statistical/optimization: Mean, MICE (van Buuren & Groothuis-Oudshoorn, 2011), SoftImpute (Mazumder et al., 2010). Trees/kNN: MissForest (Stekhoven & Bühlmann, 2012), **kNN** (Troyanskaya et al., 2001). Predictive/auto-ML and masking: HyperImpute (Jarrett et al., 2022), ReMasker (Tianyu et al., 2024). Graph-based: GRAPE (You et al., 2020). Deep generative: MIWAE (Mattei & Frellsen, 2019), GAIN (Yoon et al., 2018), normalizing flows MCFlow (Richardson et al., 2020) and MIRACLE (Trent Kyono & van der Schaar, 2021). Diffusion-based: TabCSDI (Shuhan & Nontawat, 2022), MissDiff (Yidong et al., 2023), and DiffPuter (Hengrui et al., 2025).

## 4.4 EVALUATION AND REPORTING

We evaluate imputation quality on all 11 datasets using MAE and RMSE, reporting both in-sample (train) and out-of-sample (test) metrics. Results are shown as bar charts and tables. For cross dataset comparability in the tables, we scale MAE/RMSE by 100 and report the values as percentages; the corresponding raw numbers and additional summaries are provided in the appendix. For visual clarity in the bar charts, bars are capped at 2.0; any value exceeding this cap is clipped at 2.0 and annotated with its exact value. Unless otherwise noted, all results in the results section use a 30% mask ratio.

## 4.5 IMPLEMENTATION DETAILS

Unless specified otherwise, we set the number of ODE steps to $K = 10$ and use 50 trajectory trials at inference. Training uses early stopping on the training objective. Batch sizes are set per dataset and we use SiLU activations. All experiments run on Ubuntu Linux with a single NVIDIA A100 80 GB GPU; see Appendix C for environment details. We report training time until early stopping and per sample inference time.

## 5 RESULTS

**Main quantitative results.** Across eight datasets and three missingness mechanisms (MCAR, MAR, MNAR), our Impute-MACFM attains the lowest MAE and RMSE overall, consistently outperforming or matching strong baselines in both in–sample and out–of–sample evaluations.

Table 1: Out-of-sample imputation performance under MNAR mechanism. (MAE ↓)

| Method | Dataset | | | | | | | | | | |
|---|---|---|---|---|---|---|---|---|---|---|---|
| | Letter | Gesture | Magic | Bean | Adult | Default | Shoppers | News | AHEI-S1 | AHEI-S3 | AHEI-S4 |
| Median | 76.18 | 52.11 | 68.98 | 80.03 | 54.65 | 43.79 | 37.98 | 65.28 | 82.59 | 73.88 | 80.19 |
| Mean | 77.90 | 54.00 | 71.73 | 80.00 | 60.45 | 51.79 | 53.12 | 71.72 | 85.61 | 74.93 | 81.79 |
| Most_freq | 78.00 | 88.71 | 106.20 | 140.50 | 57.54 | 50.09 | 70.37 | 128.46 | 148.59 | 119.03 | 104.53 |
| MICE | 81.42 | 64.32 | 71.17 | 21.38 | 98.70 | 59.33 | 73.52 | 70.08 | 93.05 | 82.55 | 73.88 |
| MIRACLE | 78.10 | 61.02 | 40.98 | 15.22 | 68.96 | 39.18 | 62.69 | 58.76 | 350.73 | 329.01 | 354.67 |
| Softimpute | 66.70 | 52.22 | 54.47 | 29.19 | 70.62 | 44.90 | 54.86 | 80.19 | 85.42 | 72.28 | 76.84 |
| MissForest | 58.67 | 36.49 | 46.85 | 33.34 | 61.99 | 35.94 | 47.66 | 45.35 | **67.98** | 56.41 | 66.57 |
| KNN | 63.07 | 38.11 | 53.50 | 12.77 | 71.99 | 33.79 | 45.45 | 69.65 | 101.73 | 75.32 | 96.12 |
| ReMasker | 35.70 | 33.37 | 69.33 | 12.95 | 53.72 | 31.93 | 34.03 | 40.27 | 85.49 | 66.56 | 77.48 |
| HyperImpute | 49.07 | 38.51 | 42.22 | 43.77 | 57.97 | 30.10 | 36.66 | 51.97 | 81.67 | 67.96 | 88.93 |
| GRAPE | 44.36 | 100.84 | 52.05 | 18.14 | 60.84 | 55.68 | 44.01 | 71.07 | 83.92 | 74.76 | 85.11 |
| IGRM | 135.05 | 151.92 | 110.85 | 144.78 | 105.80 | 123.63 | 121.56 | 139.77 | 124.06 | 106.08 | 108.25 |
| MIWAE | 82.18 | 57.14 | 73.65 | 84.76 | 58.83 | 46.95 | 40.33 | 69.78 | 93.06 | 81.83 | 93.95 |
| GAIN | 68.65 | 58.32 | 56.75 | 41.64 | 122.41 | 52.75 | 41.02 | 72.82 | 110.17 | 132.05 | 133.31 |
| MCFlow | 59.93 | 79.13 | 55.76 | 34.61 | 79.31 | 76.09 | 66.26 | 102.93 | 102.27 | 96.07 | 96.53 |
| TabCSDI | 77.85 | 54.62 | 71.82 | 79.71 | 62.47 | 51.44 | 54.69 | 72.57 | 95.30 | 88.31 | 90.27 |
| Missdiff | 63.28 | 398.21 | 58.90 | 58.05 | 59.65 | 371.65 | 94.76 | 204.71 | 303.72 | 185.24 | 114.48 |
| DiffPuter | 31.95 | **33.19** | 42.51 | **12.68** | 50.55 | 26.77 | 35.84 | 30.48 | 78.27 | **66.07** | 71.70 |
| **Impute-MACFM** | **31.24** | 34.24 | **40.17** | 14.01 | **45.98** | **24.56** | **32.27** | **30.47** | 71.04 | 71.12 | **65.27** |

Table 2: **Downstream classification accuracy on five UCI datasets.** We report mean accuracy across 5 independent runs. The best results are highlighted in **bold**.

| Method | ADULT | DEFAULT | MAGIC | SHOPPERS | BEAN |
|---|---|---|---|---|---|
| GAIN | 0.8137±0.003 | 0.7937±0.002 | 0.7660±0.004 | 0.8746±0.002 | 0.8606±0.005 |
| HyperImpute | 0.8273±0.002 | 0.7960±0.002 | 0.7813±0.003 | 0.8656±0.003 | 0.8783±0.004 |
| ICE | 0.8162±0.003 | 0.7990±0.002 | 0.7638±0.003 | 0.8662±0.003 | **0.9211±0.002** |
| MICE | 0.8051±0.004 | 0.7964±0.003 | 0.7471±0.005 | 0.8610±0.003 | 0.8910±0.004 |
| MIRACLE | 0.8205±0.003 | 0.7971±0.002 | 0.7778±0.003 | 0.8662±0.002 | 0.9167±0.003 |
| MIWAE | 0.8177±0.003 | 0.7931±0.002 | 0.7555±0.004 | 0.8735±0.002 | 0.7967±0.006 |
| Most-frequent | 0.8174±0.001 | 0.7952±0.002 | 0.7564±0.001 | 0.8754±0.002 | 0.7933±0.001 |
| MissForest | 0.8226±0.002 | 0.7969±0.002 | 0.7585±0.003 | 0.8692±0.002 | 0.8957±0.003 |
| SoftImpute | 0.8138±0.002 | 0.7950±0.002 | 0.7659±0.003 | 0.8708±0.002 | 0.8966±0.003 |
| MCFlow | 0.8023±0.003 | 0.7954±0.003 | 0.7326±0.005 | 0.8440±0.004 | 0.8540±0.004 |
| Remasker | 0.8146±0.003 | 0.8012±0.002 | 0.7592±0.003 | 0.8724±0.003 | 0.8902±0.003 |
| GRAPE | 0.8265±0.003 | 0.7964±0.002 | 0.7636±0.003 | 0.8746±0.003 | 0.8901±0.003 |
| DiffPuter | 0.8225±0.003 | 0.7949±0.002 | 0.7785±0.003 | 0.8635±0.003 | 0.9028±0.003 |
| **Impute-MACFM** | **0.8392±0.001** | **0.8028±0.002** | **0.7964±0.003** | **0.8802±0.001** | 0.9045±0.003 |

Summary tables in the main text report MAE and RMSE under MAR (Tables 5, 6, 7, 8) and MCAR/MNAR (Tables 9, 10, 11, 12, 13, 1, 14, 15); corresponding Histogram comparison chart appears in Figures 4, 5 and additional figures in Appendix D. Detailed per dataset results and additional analyses are provided in the Appendix tables and figures. Performance gains are most pronounced on heterogeneous datasets with mixed numeric and categorical features, reflecting the benefits of mask-aware conditioning and schedule consistent velocity learning.

**Downstream Performance.** To assess whether better imputation quality transfers to predictive performance, we run a simple downstream classification benchmark on five UCI datasets (ADULT, DEFAULT, MAGIC, SHOPPERS, BEAN). For each imputation method, we first impute the train and test splits under the same masking protocol as in the main experiments, and then train a logistic regression classifier on the imputed training data (with identical model and training hyperparameters across methods). As summarized in Table 2, Impute-MACFM achieves the best classification

Table 3: In-sample imputation time (seconds) per dataset and method (lower is better).

|              | Magic   | Letter  | Gesture | Adult  | Default | Bean    | News    | Shoppers |
|--------------|---------|---------|---------|--------|---------|---------|---------|----------|
| MissDiff     | 126.13  | 134.28  | 105.04  | 213.65 | 199.18  | 95.20   | 260.88  | 96.06    |
| TabCSDI      | 87.36   | 99.79   | 45.63   | 161.99 | 120.32  | 73.65   | 278.14  | 102.05   |
| DiffPuter    | 1017.00 | 1144.80 | 1140.00 | 847.80 | 1622.40 | 1650.00 | 1032.00 | 1512.00  |
| **Impute-MACFM** | **11.02** | **11.28** | **5.71** | **18.53** | **17.42** | **7.09** | **22.66** | **7.23** |

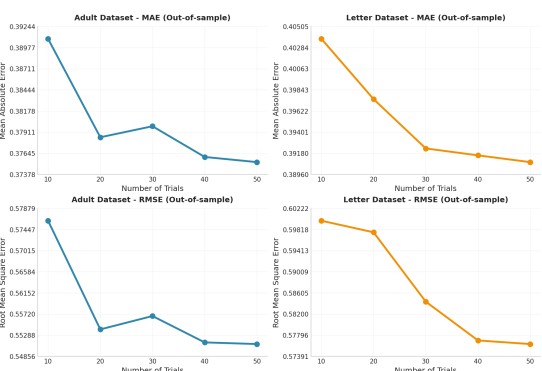

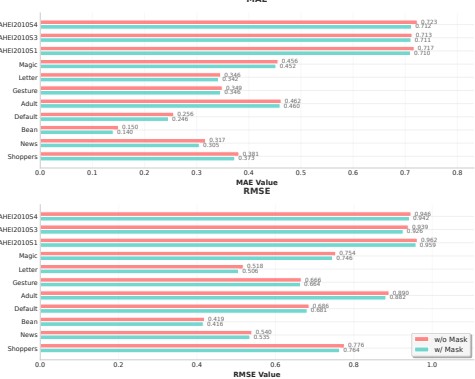

(a) **Effect of trajectory trials (Adult, Letter).** Average in-/out-of-sample MAE/RMSE across ten random masks as the number of trials increases ($10 \to 50$). Errors decrease monotonically with diminishing returns beyond $30 \sim 40$ trials; see text for relative gains. Lower is better.

(b) **Effect of mask-aware conditioning.** Out-of-sample RMSE with and without mask-aware conditioning, averaged over datasets. Mask-aware consistently improves generalization; the average relative reduction is $0.94\%$ (range $0.21\% \sim 2.20\%$). Lower is better.

Figure 2: Ablation Study on Trials and Mask-aware conditioning.

accuracy on 4 out of 5 datasets and is very close to the best method on BEAN, indicating that its gains in imputation quality translate into consistent improvements in downstream prediction rather than being purely metric-level artifacts. For the corresponding macro AUROC results, please refer to Table 16 in the appendix.

**Efficiency and scalability.** Impute-MACFM attains high quality imputations with only 10–50 ODE steps, delivering substantial inference speedups relative to diffusion baselines that typically require 100–1000 denoising steps. Training is simulation free and stable (with early stopping). Impute-MACFM is both the most accurate and the most efficient method, achieving state-of-the-art results with the lowest time.

We benchmark imputation time on eight tabular datasets against three diffusion baselines (DiffPuter, MissDiff, TabCSDI) under identical hardware. Averaged across datasets, Impute- MACFM requires $15.4$ s in-sample and $6.5$ s out-of-sample, versus DiffPuter's $1245.8$ s and $565.3$ s, yielding $125\times$ and $126\times$ speedups, respectively. Relative to lighter diffusion baselines, Impute-MACFM is $13.5\times$ faster than MissDiff and $10.7\times$ faster than TabCSDI in-sample, and $15.4\times/11.8\times$ faster out-of-sample. Per dataset runtimes are reported in Table 3.

## 5.1 ABLATION STUDIES

**Effect of trajectory trials.** We vary the number of trajectory trials $\{10, 20, 30, 40, 50\}$ and report MAE/RMSE on Adult and Letter. As shown in Fig. 2a, increasing trials consistently reduces both in-sample and out-of-sample errors with diminishing returns beyond 30 to 40. From 10 to 50 trials, Adult improves by 5.9% MAE and 6.1% RMSE in-sample, and 4.0% MAE and 4.4% RMSE out-of-sample; Letter improves by 3.5% MAE and 9.4% RMSE in-sample, and 3.2% MAE and 3.9% RMSE out-of-sample.

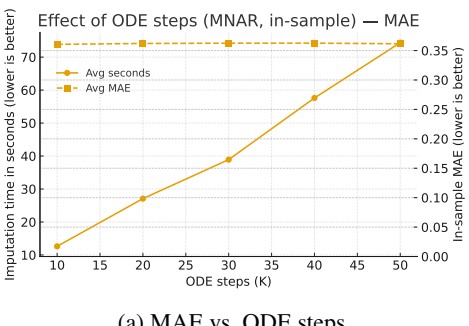

(a) MAE vs. ODE steps.

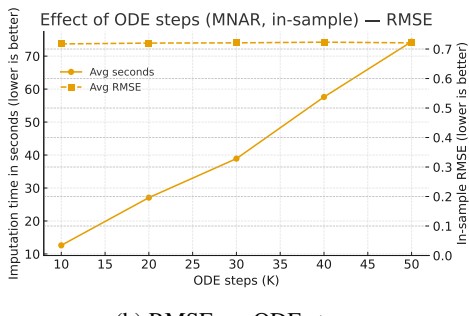

(b) RMSE vs. ODE steps.

Figure 3: Each point averages over eight datasets. Runtime increases roughly linearly with $K$, while MAE/RMSE remain essentially unchanged across $K \in \{10, 20, 30, 40, 50\}$. Thus $K=10$ attains the same quality at a fraction of the compute (lower is better).

**Effect of Mask-aware conditioning**  We ablate the mask-aware objective by disabling it and re-running inference. Across eleven datasets, enabling mask-aware consistently lowers out-of-sample RMSE (Figure 2), yielding an average relative reduction of 0.94% (range: 0.21%–2.20%). The largest gains are observed on *Letter* ($0.5175 \rightarrow 0.5061$, $-2.20\%$), *Shoppers* ($0.7760 \rightarrow 0.7642$, $-1.52\%$), *AHEI2010S3* ($0.9386 \rightarrow 0.9255$, $-1.39\%$), and *Magic* ($0.7538 \rightarrow 0.7460$, $-1.03\%$). Improvements also hold on *Adult* ($0.8899 \rightarrow 0.8819$, $-0.89\%$) and *News* ($0.5398 \rightarrow 0.5351$, $-0.86\%$), indicating that mask-aware conditioning strengthens constraint handling and generalization in out-of-sample settings.

**Effect of ODE steps**  Under MNAR and in-sample evaluation, figure 3 shows that runtime grows nearly linearly with the number of ODE steps while performance is essentially unchanged. Using $K=10$ reduces the average imputation time by about $6\times$ (from $\approx 74$s at $K=50$ to $\approx 13$s) with an almost flat MAE curve; RMSE exhibits the same stability. We therefore adopt $K=10$ as the default, which provides the same quality at a fraction of the computational cost.

## 6 CONCLUSION

We present Impute-MACFM, a mask-aware conditional flow-matching framework for heterogeneous tabular and longitudinal imputation. It combines (i) a *schedule-consistent* velocity target, (ii) a *mask-aware* objective that updates only missing entries while penalizing motion on conditioning entries, and (iii) a constraint-preserving ODE solver (projection + Heun). Training adds a local-consistency regularizer and light numeric noise; inference attains high quality with few ODE steps and supports multi-trajectory aggregation. Across 11 datasets (8 public + 3 NIH AHEI), Impute-MACFM achieves state-of-the-art MAE/RMSE, particularly on mixed numeric/categorical data. Average runtime is $15.4$ s (in-sample) and $6.5$ s (out-of-sample)—$125 \times /126\times$ faster than DiffPuter, and $13.7 \times /10.7\times$ (in-sample) and $15.4 \times /11.8\times$ (out-of-sample) faster than MissDiff/TabCSDI. Few-step ablations show $K=10$ matches $K=50$ with $\sim 5\times$ lower wall time; mask-aware ablations reduce out-of-sample RMSE (avg $-0.94\%$, up to $-2.20\%$). The same advantages hold on AHEI Dataset.

## ETHICS STATEMENT

We affirm compliance with the ICLR Code of Ethics. This work investigates methods for missing-value imputation on tabular data with the goal of improving data quality and evaluation practice; it does not involve high-risk applications.

**Public datasets.** All public datasets were obtained from the UCI Machine Learning Repository or Kaggle under their respective terms. We do not redistribute any restricted copies and follow the original licenses.

**NIH datasets.** The NIH datasets (AHEI_2010_S1/S3/S4) are fully de-identified and used under data use agreements. The data contain no direct identifiers (e.g., names, addresses, exact dates of birth) and no linkage keys are available to the authors; records are keyed only by random study IDs. To preserve double-blind review, we include no metadata that could reveal author or institutional identities in this submission; IRB/DUA identifiers will be disclosed in the camera-ready version if required.

**Potential risks and misuse.** Imputation may introduce bias if treated as ground truth. We mitigate this risk by reporting results across repeated random masks, evaluating both in-sample and out-of-sample settings, comparing against diverse baselines, and documenting limitations. We caution against causal interpretation without appropriate identification assumptions.

**Fairness and bias.** Because data distributions differ across domains, we recommend re-evaluation of imputation performance before deployment in sensitive settings, especially when sensitive attributes are present.

**Conflicts of interest.** The authors declare no competing interests.

## REPRODUCIBILITY STATEMENT

We release an *anonymized* GitHub repository with code for end-to-end reproduction of all MAE/RMSE results (in- and out-of-sample): `https://github.com/code-repo-papers/Impute-MACFM`. The repository contains scripts for data downloading, preprocessing, training, and evaluation; it will be de-anonymized after the review process.

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

# A DEFINITION OF SYMBOLS AND EQUATION PROOFS

## A.1 SYMBOLS AND NOTATION

**Symbols**

| | | |
|---|---|---|
| $D$ | | Number of features |
| $\mathbf{X} \in \mathbb{R}^D$ | | Input vector with $D$ features |
| $\mathbf{x}_t$ | | State along the conditional path at time $t$ |
| $M_{\text{obs}}, M_{\text{cond}}, M_{\text{tgt}} \in \{0,1\}^D$ | | Masks for observed, conditional, and target features (disjoint). $M_{\text{obs}} + M_{\text{cond}} + M_{\text{tgt}} = \mathbf{1}$ |
| $s(t)$ | | Schedule function $[0,1] \to [0,1]$ with $s(0) = 0$ and $s(1) = 1$ |
| $s'(t)$ | | Derivative of the schedule |
| $\boldsymbol{\varepsilon}$ | | Standard normal noise vector |
| $\mathbf{v}_\theta(\mathbf{x}_t, t)$ | | Predicted velocity field |
| $\mathbf{v}_\star(\mathbf{x}_t, t)$ | | Target velocity $M_{\text{tgt}} \odot s'(t)(\mathbf{X} - \boldsymbol{\varepsilon})$ |
| $\odot$ | | Elementwise Hadamard product |
| $\| \cdot \|_2$ | | Euclidean norm |
| $\mathbb{E}[\cdot]$ | | Expectation over variables indicated by context |
| $\mathcal{L}_{\text{FM}}$ | | Flow matching loss computed on $M_{\text{tgt}}$ |
| $\mathcal{R}_{\text{stab}}$ | | Stability regularizer on $M_{\text{cond}}$ |
| $\mathcal{R}_{\text{cons}}$ | | Consistency regularizer on $M_{\text{tgt}}$ |
| $\sigma_{\text{in}}$ | | Input noise scale for numeric features |
| $\eta_{\text{cons}}$ | | Magnitude of the consistency perturbation |
| $n_{\text{num}}$ | | Number of numeric features |
| $M_{\text{num}}$ | | Numeric feature mask in $\{0,1\}^D$ |
| $K$ | | Number of integration steps |
| $t_k$ | | Discrete time at step $k$ with $t_k = k/K$ |
| $s_k$ | | Grid point in schedule space with $s_k = s(t_k)$ |
| $\Delta t, \ \Delta s_k$ | | Step sizes in time or schedule space |

## A.2 DERIVATION OF TARGET VELOCITY FIELD

Starting from the path definition in equation 1:

$$\mathbf{x}_t = M_{\text{cond}} \odot \mathbf{X} + M_{\text{tgt}} \odot \big(s(t)\mathbf{X} + (1 - s(t))\boldsymbol{\varepsilon}\big) \tag{12}$$

Taking the time derivative with respect to $t$:

$$\frac{\partial \mathbf{x}_t}{\partial t} = \frac{\partial}{\partial t} \big[ M_{\text{cond}} \odot \mathbf{X} + M_{\text{tgt}} \odot \big(s(t)\mathbf{X} + (1 - s(t))\boldsymbol{\varepsilon}\big) \big] \tag{13}$$

$$= 0 + M_{\text{tgt}} \odot [s'(t)\mathbf{X} - s'(t)\boldsymbol{\varepsilon}] \tag{14}$$

$$= M_{\text{tgt}} \odot s'(t)(\mathbf{X} - \boldsymbol{\varepsilon}) \tag{15}$$

To obtain the velocity in schedule-space, we use the chain rule:

$$\mathbf{v}_\star^{(s)}(\mathbf{x}_t, t) = \frac{\partial \mathbf{x}_t}{\partial s} = \frac{\partial \mathbf{x}_t / \partial t}{\partial s / \partial t} = \frac{M_{\text{tgt}} \odot s'(t)(\mathbf{X} - \boldsymbol{\varepsilon})}{s'(t)} \tag{16}$$

$$= M_{\text{tgt}} \odot (\mathbf{X} - \boldsymbol{\varepsilon}) \tag{17}$$

## A.3 EQUIVALENCE OF TRAINING OBJECTIVES

**Proposition 1.** The following two objectives are mathematically equivalent:

**Schedule-domain objective:**

$$\mathcal{L}_{\text{FM}}^{(s)}(\theta) = \mathbb{E}_{\mathbf{X},\boldsymbol{\varepsilon},t}\left[\left\|\left(\mathbf{v}_\theta(\mathbf{x}_t, t) - \mathbf{v}_\star^{(s)}(\mathbf{x}_t, t)\right) \odot M_{\text{tgt}}\right\|_2^2\right] \tag{18}$$

**Time-domain objective:**

$$\mathcal{L}_{\text{FM}}^{(t)}(\theta) = \mathbb{E}_{\mathbf{X},\boldsymbol{\varepsilon},t}\left[\left\|\left(\mathbf{v}_\theta(\mathbf{x}_t, t) - \mathbf{v}_\star^{(t)}(\mathbf{x}_t, t)\right) \odot M_{\text{tgt}}\right\|_2^2\right] \tag{19}$$

**Proof:** The equivalence follows from the relationship between velocities in different parameterizations. If we interpret $\mathbf{v}_\theta$ as approximating $\frac{\partial \mathbf{x}}{\partial s}$, we use objective equation 18. If we interpret it as approximating $\frac{\partial \mathbf{x}}{\partial t}$, we must scale by $s'(t)$ as in objective equation 19. The optimal solutions satisfy:

$$\mathbf{v}_\theta^{*(s)}(\mathbf{x}_t, t) = \mathbf{v}_\star^{(s)}(\mathbf{x}_t, t) = M_{\text{tgt}} \odot (\mathbf{X} - \boldsymbol{\varepsilon}) \tag{20}$$

$$\mathbf{v}_\theta^{*(t)}(\mathbf{x}_t, t) = \mathbf{v}_\star^{(t)}(\mathbf{x}_t, t) = s'(t) \cdot M_{\text{tgt}} \odot (\mathbf{X} - \boldsymbol{\varepsilon}) \tag{21}$$

## A.4 LOSS WITH MASKING AND NORMALIZATION

We fit the velocity only on target dimensions. Let $\text{den}_{\text{tgt}} = \sum M_{\text{tgt}} + \epsilon$ denote the number of target entries with a small constant $\epsilon > 0$. The time domain flow matching loss used in practice is

$$\mathcal{L}_{\text{FM}}^{(t)}(\theta) = \mathbb{E}_{\mathbf{X},\boldsymbol{\varepsilon},t}\left[\frac{\left\|\left(\mathbf{v}_\theta(\mathbf{x}_t, t) - M_{\text{tgt}} \odot s'(t)(\mathbf{X} - \boldsymbol{\varepsilon})\right) \odot M_{\text{tgt}}\right\|_2^2}{\text{den}_{\text{tgt}}}\right]. \tag{22}$$

## A.5 STABILITY AND CONSISTENCY REGULARIZERS

To discourage movement on condition dimensions we add a stability term with $\text{den}_{\text{cond}} = \sum M_{\text{cond}} + \epsilon$:

$$\mathcal{R}_{\text{stab}}(\theta) = \mathbb{E}\left[\frac{\left\|\mathbf{v}_\theta(\mathbf{x}_t, t) \odot M_{\text{cond}}\right\|_2^2}{\text{den}_{\text{cond}}}\right]. \tag{23}$$

We further promote local smoothness on target dimensions through a consistency term with $\boldsymbol{\xi} \sim \mathcal{N}(0, \mathbf{I})$ and a perturbation that decays with $(1 - s(t))$:

$$\mathcal{R}_{\text{cons}}(\theta) = \mathbb{E}\left[\frac{\left\|\left(\mathbf{v}_\theta(\mathbf{x}_t + \eta_{\text{cons}}(1 - s(t))\boldsymbol{\xi}, t) - \mathbf{v}_\theta(\mathbf{x}_t, t)\right) \odot M_{\text{tgt}}\right\|_2^2}{\text{den}_{\text{tgt}}}\right]. \tag{24}$$

The complete objective matches the Method section:

$$\mathcal{L}_{\text{total}}(\theta) = \mathcal{L}_{\text{FM}}^{(t)}(\theta) + \lambda_{\text{stab}}\,\mathcal{R}_{\text{stab}}(\theta) + \lambda_{\text{cons}}\,\mathcal{R}_{\text{cons}}(\theta). \tag{25}$$

## A.6 NUMERIC ONLY INPUT NOISE

We inject small input noise on observed numeric features with magnitude that decays with $(1 - s(t))$. Let the first $n_{\text{num}}$ dimensions be numeric. Define a numeric mask $M_{\text{num}} \in \{0, 1\}^D$ with ones on the first $n_{\text{num}}$ positions and zeros elsewhere. The augmentation used in training is

$$\mathbf{X} \leftarrow \mathbf{X} + \sigma_{\text{in}}(1 - s(t))\boldsymbol{\xi} \odot M_{\text{obs}} \odot M_{\text{num}}, \quad \boldsymbol{\xi} \sim \mathcal{N}(0, \mathbf{I}). \tag{26}$$

## A.7 HEUN'S METHOD WITH PROJECTION

We employ a second-order Heun predictor-corrector scheme. Given a partition $0 = s_0 < s_1 < \ldots < s_K = 1$ with $\Delta s_k = s_{k+1} - s_k$:

**Initialization:**

$$\mathbf{x}^{(0)} = M_{\text{obs}} \odot \mathbf{X} + M_{\text{cond}} \odot \mathbf{X} + M_{\text{tgt}} \odot \varepsilon \tag{27}$$

**For** $k = 0, 1, \ldots, K-1$**:**

*Step 1: Predictor (Euler)*

$$\tilde{\mathbf{x}}^{(k+1)} = \mathbf{x}^{(k)} + \Delta s_k \cdot \mathbf{v}_\theta(\mathbf{x}^{(k)}, t_k) \tag{28}$$

*Step 2: Project to constraints*

$$\tilde{\mathbf{x}}^{(k+1)} \leftarrow \tilde{\mathbf{x}}^{(k+1)} \odot M_{\text{tgt}} + \mathbf{X} \odot M_{\text{obs}} \tag{29}$$

*Step 3: Corrector (Heun)*

$$\mathbf{x}^{(k+1)} = \mathbf{x}^{(k)} + \frac{\Delta s_k}{2} \left( \mathbf{v}_\theta(\mathbf{x}^{(k)}, t_k) + \mathbf{v}_\theta(\tilde{\mathbf{x}}^{(k+1)}, t_{k+1}) \right) \tag{30}$$

*Step 4: Final projection*

$$\mathbf{x}^{(k+1)} \leftarrow \mathbf{x}^{(k+1)} \odot M_{\text{tgt}} + \mathbf{X} \odot M_{\text{obs}} \tag{31}$$

where $t_k = s^{-1}(s_k)$ for the chosen schedule function.

## A.8 SCHEDULE FUNCTIONS

**Power Schedule:**

$$s(t) = t^\gamma, \quad \gamma \in [1, 3] \tag{32}$$

$$s'(t) = \gamma t^{\gamma-1} \tag{33}$$

$$s^{-1}(s) = s^{1/\gamma} \tag{34}$$

**Cosine Schedule:**

$$s(t) = \frac{1}{2} \left( 1 - \cos(\pi t) \right) \tag{35}$$

$$s'(t) = \frac{\pi}{2} \sin(\pi t) \tag{36}$$

$$s^{-1}(s) = \frac{1}{\pi} \arccos(1 - 2s) \tag{37}$$

## A.9 ADAPTIVE STEP SIZE SELECTION (OPTIONAL)

For non-uniform grids in schedule space, one may concentrate steps near $s = 1$:

$$s_k = \left( \frac{k}{K} \right)^\beta, \quad k = 0, 1, \ldots, K \tag{38}$$

where $\beta > 1$ creates denser sampling near the data distribution. In our default experiments, we use uniform time steps $t_k = k/K$.

## B DATASETS INFORMATION

Table 4 summarizes the datasets used in our imputation benchmarks. Instance counts follow the official releases. Numeric and categorical counts follow our preprocessed schema described below. We use a random 70% and 30% split for training and test sets, respectively, with a fixed seed. For each dataset, we generate ten masks per mechanism under MCAR, MAR, and MNAR at a nominal missing rate of 0.3.

### B.1 EIGHT PUBLIC DATASETS

We evaluate on eight real world datasets drawn from the UCI Machine Learning Repository (and mirrors), spanning four with only continuous features and four with mixed continuous and discrete features.

Table 4: Statistics of datasets

| Dataset | Rows | Columns | Train (In-sample) | # Test (Out-of-sample) |
|---|---|---|---|---|
| *Letter* | 20,000 | 16 | 14,000 | 6,000 |
| *Gesture* | 9,522 | 32 | 6,665 | 2,857 |
| *Magic* | 19,020 | 10 | 13,314 | 5,706 |
| *Bean* | 13,610 | 17 | 9,527 | 4,083 |
| *Adult* | 32,561 | 14 | 22,792 | 9,769 |
| *Default* | 30,000 | 24 | 21,000 | 9,000 |
| *Shoppers* | 12,330 | 17 | 8,631 | 3,699 |
| *News* | 39,644 | 47 | 27,790 | 11,894 |
| *AHEI2010-S1* | 227 | 20 | 182 | 45 |
| *AHEI2010-S3* | 201 | 24 | 156 | 45 |
| *AHEI2010-S4* | 220 | 26 | 175 | 45 |

**Adult (Census Income; mixed).**   The Adult dataset predicts whether an individual's annual income exceeds \$50K using demographic attributes extracted from the 1994 U.S. Census. It is a binary classification benchmark with both categorical and integer-valued features.[1]

**Default of Credit Card Clients (mixed).**   This dataset contains Taiwanese credit-card client records (demographics, billing, payment status, etc.) and is commonly used for default-risk prediction.[2]

**MAGIC Gamma Telescope (continuous-only).**   MC-generated feature vectors simulate registrations of high-energy gamma particles recorded by a ground-based atmospheric Cherenkov telescope; the task is to distinguish gamma events from hadronic background.[3]

**Online Shoppers Purchasing Intention (mixed).**   Session-level e-commerce behavior (technical & behavioral features) for predicting purchase intention; collected and documented with a standardized protocol.[4]

**Online News Popularity (mixed).**   Article-level features from Mashable over two years; the goal is to predict the number of social-network shares (often framed as a regression target).[5]

**Gesture Phase Segmentation (continuous-only).**   Features are extracted from 7 recorded gesture videos; for each video, position streams and processed velocity/acceleration files are provided to study gesture phase segmentation.[6]

**Letter Recognition (continuous-only).**   A character-recognition benchmark in which 16 numeric attributes summarize letter images (20,000 instances across 26 classes).[7]

**Dry Bean (continuous-only).**   A computer-vision–derived tabular dataset with morphometric measurements for 13,611 dry-bean images across seven cultivars; used for multiclass classification.[8]

### B.2   3 PRIVATE NIH DATASETS (AHEI2010)

**Harmonization note.**   In the harmonized dataset, these trials are labeled as `study=1`, `study=3`, and `study=4`.

---

[1] https://archive.ics.uci.edu/dataset/2/adult

[2] https://archive.ics.uci.edu/dataset/350/default+of+credit+card+clients

[3] https://archive.ics.uci.edu/ml/datasets/magic+gamma+telescope

[4] https://archive.ics.uci.edu/ml/datasets/Online+Shoppers+Purchasing+Intention+Dataset

[5] https://archive.ics.uci.edu/ml/datasets/online+news+popularity

[6] https://archive.ics.uci.edu/dataset/302/gesture+phase+segmentation

[7] https://archive.ics.uci.edu/ml/datasets/letter+recognition

[8] https://archive.ics.uci.edu/ml/datasets/dry+bean+dataset

Table 5: In-sample imputation performance under MAR mechanism. (MAE ↓)

| Method | Dataset | | | | | | | | | | |
|---|---|---|---|---|---|---|---|---|---|---|---|
| | Letter | Gesture | Magic | Bean | Adult | Default | Shoppers | News | AHEI-S1 | AHEI-S3 | AHEI-S4 |
| Median | 75.89 | 54.48 | 72.34 | 86.35 | 45.66 | 54.49 | 43.15 | 57.51 | 77.30 | 77.05 | 72.47 |
| Mean | 78.74 | 57.76 | 73.91 | 85.11 | 56.12 | 64.02 | 57.68 | 64.28 | 79.57 | 80.83 | 74.95 |
| Most_freq | 76.70 | 99.66 | 99.40 | 197.12 | 47.19 | 59.99 | 85.39 | 111.99 | 130.96 | 111.32 | 113.96 |
| EM | 54.80 | 36.25 | 46.71 | 11.83 | 52.24 | 32.01 | 39.51 | 40.57 | 67.64 | 75.34 | 69.11 |
| MICE | 80.70 | 63.60 | 71.39 | 17.29 | 97.47 | 56.00 | 74.35 | 64.89 | 91.60 | 84.56 | 79.22 |
| MIRACLE | 66.75 | 60.85 | 47.74 | 20.90 | 53.68 | 43.22 | 59.35 | 44.71 | 332.78 | 376.25 | 386.90 |
| Softimpute | 63.89 | 59.95 | 55.81 | 33.67 | 56.56 | 49.47 | 58.10 | 64.85 | 78.13 | 66.38 | 69.88 |
| MissForest | 59.54 | 35.21 | 44.59 | 28.78 | 64.22 | 38.33 | 36.10 | 38.22 | 65.45 | 55.96 | 67.35 |
| KNN | 59.62 | 39.72 | 51.57 | 13.59 | 82.95 | 38.45 | 47.69 | 44.59 | 89.79 | 67.53 | 76.97 |
| ReMasker | 34.12 | 32.21 | 39.95 | 12.66 | 39.74 | 36.54 | 34.08 | 30.63 | 67.30 | 59.36 | 70.20 |
| HyperImpute | 53.09 | 37.95 | 47.05 | 55.10 | 56.37 | 28.56 | – | – | 66.25 | 56.44 | 61.99 |
| GRAPE | 42.01 | 35.44 | 37.59 | 12.86 | 55.28 | 47.88 | 46.31 | 50.00 | 73.71 | 69.52 | 72.38 |
| IGRM | 41.55 | 40.62 | 37.61 | 13.08 | 54.05 | 56.89 | 54.38 | 49.94 | 76.96 | 69.60 | 73.90 |
| MIWAE | 83.42 | 59.48 | 76.80 | 89.24 | 50.02 | 58.63 | 45.05 | 61.64 | 85.65 | 85.40 | 82.94 |
| GAIN | 78.95 | 90.76 | 61.60 | 53.37 | 132.32 | 93.35 | 46.03 | 55.55 | 75.74 | 68.03 | 75.99 |
| MCFlow | 59.38 | 61.49 | 54.12 | 32.31 | 61.27 | 69.61 | 75.95 | 52.68 | 100.62 | 101.48 | 96.69 |
| TabCSDI | 76.62 | 57.36 | 73.24 | 79.35 | 55.88 | 59.90 | 57.02 | 63.79 | 88.82 | 85.14 | 78.18 |
| Missdiff | 46.65 | 51.97 | 96.80 | 72.39 | 57.79 | 69.18 | 44.60 | 47.95 | 113.71 | 100.50 | 156.63 |
| DiffPuter | 39.53 | 40.72 | 41.73 | 35.04 | 40.08 | 48.53 | 45.46 | 35.73 | 77.54 | 72.47 | 66.44 |
| **Impute-MACFM** | 25.59 | 24.22 | 32.13 | 15.12 | 43.99 | 27.75 | 34.52 | 29.13 | 72.27 | 52.04 | 53.59 |

## C  EXPERIMENTAL SETUP

All experiments ran on Ubuntu 22.04.5 LTS (Linux 5.15). The host has 2× Intel Xeon Platinum 8468 CPUs (192 threads total) and 8 NVIDIA A100 80 GB GPUs. The NVIDIA driver is compatible with CUDA 12.3; the user-space CUDA runtime is 11.8. Python 3.10 was used. Unless otherwise stated, both training and inference used one GPU.

## D  SUPPLEMENTARY RESULTS TABLES AND FIGURES

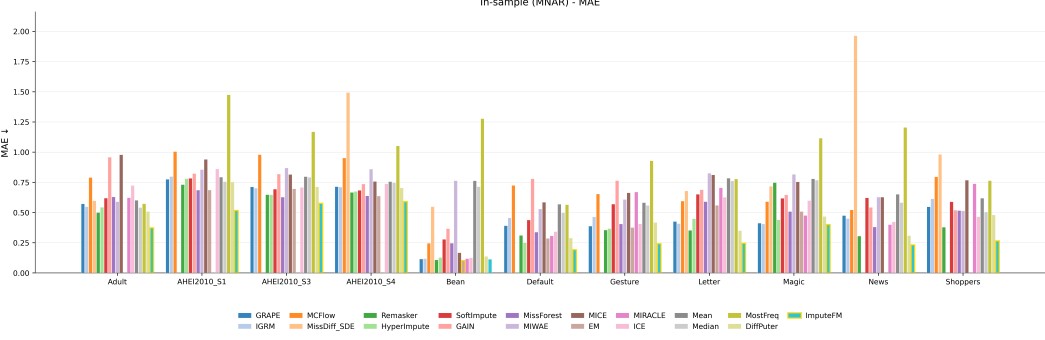

Figure 4: MNAR, In-sample imputation performance on MAE score (lower is better)

Table 6: Out-of-sample imputation performance under MAR mechanism. (MAE ↓)

| Method | Dataset | | | | | | | | | | |
|---|---|---|---|---|---|---|---|---|---|---|---|
| | Letter | Gesture | Magic | Bean | Adult | Default | Shoppers | News | AHEI-S1 | AHEI-S3 | AHEI-S4 |
| Median | 75.61 | 54.02 | 67.80 | 75.08 | 47.25 | 47.20 | 43.07 | 59.58 | 74.00 | 79.37 | 74.83 |
| Mean | 78.45 | 56.10 | 68.98 | 76.91 | 57.84 | 54.20 | 58.56 | 64.66 | 78.26 | 79.36 | 77.23 |
| Most_freq | 76.73 | 85.89 | 87.94 | 182.41 | 46.96 | 52.97 | 44.86 | 135.58 | 167.09 | 116.26 | 106.17 |
| MICE | 80.91 | 67.15 | 72.89 | 18.27 | 103.19 | 60.62 | 77.69 | 98.94 | 101.14 | 86.35 | 85.75 |
| MIRACLE | 67.50 | 67.39 | 43.28 | 20.59 | – | 54.37 | 80.52 | 55.44 | 264.48 | 341.41 | 358.36 |
| Softimpute | 65.46 | 57.76 | 53.54 | 31.32 | 58.07 | 42.77 | 55.83 | 64.67 | 77.11 | 76.58 | 74.14 |
| MissForest | 57.01 | 38.63 | 43.50 | 23.96 | 62.42 | 30.27 | 44.96 | 39.11 | 61.97 | 67.15 | 68.53 |
| KNN | 64.40 | 42.10 | 49.20 | 14.51 | – | 34.28 | 48.13 | 48.86 | 86.24 | 81.53 | 91.91 |
| ReMasker | 33.87 | 40.11 | 40.51 | 15.16 | 44.03 | 31.45 | 37.03 | 31.48 | 79.84 | 72.21 | 74.26 |
| HyperImpute | 56.20 | 41.75 | 41.73 | 15.06 | – | 40.51 | 43.47 | 49.87 | 74.61 | 73.92 | 87.53 |
| GRAPE | 42.50 | 83.86 | 46.85 | 18.83 | 57.45 | 56.42 | 46.84 | 82.17 | 81.01 | 87.92 | 76.20 |
| IGRM | 108.41 | 126.05 | 122.31 | 122.04 | 114.35 | 118.20 | 111.99 | 118.48 | 116.40 | 112.45 | 101.43 |
| MIWAE | 82.61 | 59.00 | 72.16 | 79.98 | – | 49.93 | 44.94 | 63.88 | 83.84 | 87.39 | 87.59 |
| GAIN | 65.37 | 86.47 | 66.43 | 44.84 | 75.75 | 115.45 | 38.65 | 68.96 | 114.70 | 105.45 | 144.55 |
| MCFlow | 59.29 | 81.99 | 61.25 | 37.35 | 74.44 | 77.31 | 76.89 | 92.18 | 100.99 | 97.00 | 96.10 |
| TabCSDI | 77.28 | 56.57 | 68.65 | 74.59 | 57.63 | 53.09 | 58.81 | 64.73 | 90.14 | 90.89 | 80.10 |
| Missdiff | 46.62 | 63.29 | 86.85 | 42.74 | 62.54 | 148.40 | 50.53 | 49.53 | 210.19 | 120.18 | 556.20 |
| DiffPuter | 33.83 | 39.29 | 40.96 | 37.11 | 40.94 | 29.70 | 41.28 | 30.39 | 75.37 | 72.01 | 72.32 |
| **Impute-MACFM** | 40.66 | 32.44 | 45.72 | 15.34 | 48.35 | 32.01 | 43.14 | 29.87 | 71.03 | 71.12 | 71.19 |

Table 7: In-sample imputation performance under MAR mechanism. (RMSE ↓)

| Method | Dataset | | | | | | | | | | |
|---|---|---|---|---|---|---|---|---|---|---|---|
| | Letter | Gesture | Magic | Bean | Adult | Default | Shoppers | News | AHEI-S1 | AHEI-S3 | AHEI-S4 |
| Median | 102.51 | 105.03 | 103.81 | 121.76 | 89.80 | 111.10 | 108.51 | 100.40 | 102.92 | 111.63 | 93.68 |
| Mean | 102.21 | 102.55 | 101.10 | 118.58 | 90.85 | 110.76 | 104.06 | 96.82 | 101.10 | 104.90 | 94.75 |
| Most_freq | 103.90 | 165.17 | 131.87 | 261.19 | 93.58 | 122.26 | 168.03 | 182.05 | 169.51 | 153.13 | 153.57 |
| EM | 74.89 | 70.84 | 75.87 | 31.69 | 83.35 | 81.25 | 77.17 | 69.61 | 91.20 | 109.26 | 98.59 |
| MICE | 105.38 | 93.23 | 102.35 | 42.17 | 129.71 | 99.94 | 107.71 | 94.93 | 118.76 | 116.82 | 109.63 |
| MIRACLE | 96.50 | 105.77 | 81.53 | 62.24 | 100.51 | 102.28 | 119.16 | 84.96 | 361.19 | 408.06 | 419.22 |
| Softimpute | 86.22 | 103.54 | 87.97 | 53.47 | 90.29 | 95.90 | 102.45 | 103.34 | 99.49 | 93.48 | 92.39 |
| MissForest | 81.92 | 72.09 | 74.43 | 49.39 | 126.61 | 94.82 | 83.07 | 67.27 | 90.32 | 88.67 | 90.40 |
| KNN | 80.83 | 74.38 | 82.49 | 33.65 | 142.99 | 86.67 | 91.43 | 73.49 | 122.72 | 106.94 | 104.96 |
| ReMasker | 48.57 | 66.40 | 66.45 | 33.95 | 76.66 | 82.44 | 75.53 | 58.34 | 91.79 | 89.63 | 92.41 |
| HyperImpute | 71.71 | 71.39 | 73.28 | 66.29 | 103.00 | 75.25 | – | – | 88.83 | 92.72 | 81.67 |
| GRAPE | 57.60 | 67.87 | 62.88 | 32.09 | 99.17 | 99.49 | 91.88 | 79.11 | 98.47 | 103.46 | 98.30 |
| IGRM | 57.13 | 75.03 | 63.51 | 33.58 | 88.41 | 101.32 | 97.64 | 77.57 | 104.76 | 104.84 | 99.12 |
| MIWAE | 107.23 | 106.50 | 107.26 | 124.13 | 92.70 | 112.86 | 108.86 | 102.22 | 110.27 | 113.70 | 105.75 |
| GAIN | 104.17 | 136.29 | 87.14 | 78.23 | 200.63 | 147.56 | 107.68 | 88.05 | 101.34 | 102.23 | 103.56 |
| MCFlow | 79.29 | 105.57 | 85.46 | 52.80 | 101.30 | 113.25 | 120.68 | 81.23 | 128.11 | 131.71 | 122.55 |
| TabCSDI | 100.03 | 102.36 | 100.48 | 111.16 | 90.42 | 106.87 | 102.34 | 97.23 | 111.24 | 111.55 | 104.24 |
| Missdiff | 65.42 | 167.16 | 208.56 | 119.95 | 92.62 | 221.15 | 110.78 | 195.22 | 211.54 | 138.90 | 329.08 |
| DiffPuter | 58.77 | 93.56 | 70.90 | 107.31 | 78.84 | 187.69 | 120.76 | 65.06 | 101.71 | 102.65 | 87.63 |
| **Impute-MACFM** | 37.27 | 46.64 | 52.10 | 36.76 | 82.10 | 78.24 | 77.82 | 49.63 | 94.59 | 73.05 | 78.41 |

Table 8: Out-of-sample imputation performance under MAR mechanism. (RMSE ↓)

| | | | | | | | Dataset | | | | |
| Method | Letter | Gesture | Magic | Bean | Adult | Default | Shoppers | News | AHEI-S1 | AHEI-S3 | AHEI-S4 |
|---|---|---|---|---|---|---|---|---|---|---|---|
| Median | 103.96 | 101.69 | 95.57 | 112.35 | 99.82 | 104.71 | 104.01 | 99.23 | 99.49 | 110.13 | 99.37 |
| Mean | 102.29 | 100.05 | 92.91 | 111.00 | – | 100.39 | 98.96 | 94.53 | 97.34 | 100.82 | 99.87 |
| Most_freq | 105.94 | 146.65 | 121.86 | 257.55 | 101.05 | 115.46 | 108.93 | 203.37 | 202.83 | 157.74 | 143.12 |
| MICE | 105.90 | 95.91 | 99.33 | 42.87 | 140.18 | 97.85 | 107.90 | 1229.84 | 126.42 | 113.08 | 116.80 |
| MIRACLE | 98.85 | 122.79 | 65.78 | 48.85 | – | 120.51 | 134.79 | 438.78 | 290.76 | 373.12 | 418.89 |
| Softimpute | 88.08 | 102.18 | 73.12 | 52.95 | 99.84 | 102.16 | 91.27 | 163.28 | 96.74 | 96.21 | 96.33 |
| MissForest | 79.52 | 76.80 | 64.77 | 42.13 | 132.84 | 77.35 | 83.13 | 66.31 | 85.89 | 96.32 | 92.23 |
| KNN | 87.10 | 76.22 | 72.50 | 33.03 | – | 83.32 | 88.20 | 89.42 | 111.10 | 111.74 | 123.46 |
| ReMasker | 48.44 | 78.73 | 61.49 | 34.67 | 89.62 | 75.02 | 75.07 | 56.76 | 105.05 | 99.95 | 102.56 |
| HyperImpute | 75.02 | 75.73 | 63.70 | 36.45 | – | 79.33 | 84.56 | 81.25 | 96.55 | 104.52 | 116.63 |
| GRAPE | 58.96 | 115.07 | 68.28 | 36.73 | 101.70 | 101.26 | 86.04 | 113.81 | 108.17 | 115.30 | 100.94 |
| IGRM | 138.68 | 152.88 | 153.40 | 150.71 | 147.40 | 157.22 | 153.03 | 146.82 | 138.46 | 137.62 | 128.26 |
| MIWAE | 107.36 | 103.31 | 99.33 | 116.16 | – | 105.63 | 104.40 | 100.12 | 111.22 | 112.90 | 114.01 |
| GAIN | 87.41 | 139.22 | 89.75 | 71.15 | 127.53 | 224.58 | 93.15 | 108.62 | 171.89 | 142.05 | 186.23 |
| MCFlow | 80.75 | 124.91 | 87.72 | 61.53 | 123.96 | 116.80 | 125.38 | 125.26 | 130.63 | 126.98 | 122.69 |
| TabCSDI | 101.45 | 100.41 | 93.27 | 106.52 | 100.47 | 100.31 | 97.75 | 95.03 | 116.73 | 114.70 | 107.26 |
| Missdiff | 65.74 | 379.56 | 158.12 | 76.06 | 106.39 | 257.77 | 319.72 | 247.95 | 612.94 | 164.50 | 2371.11 |
| DiffPuter | 52.58 | 85.88 | 64.13 | 120.35 | 87.47 | 112.27 | 103.8 | 56.42 | 101.83 | 100.92 | 99.59 |
| **Impute-MACFM** | 58.59 | 65.52 | 68.38 | 37.58 | 90.87 | 81.59 | 84.42 | 59.54 | 94.60 | 84.64 | 103.02 |

Table 9: In-sample imputation performance under MCAR mechanism. (MAE ↓)

| | | | | | | | Dataset | | | | |
| Method | Letter | Gesture | Magic | Bean | Adult | Default | Shoppers | News | AHEI-S1 | AHEI-S3 | AHEI-S4 |
|---|---|---|---|---|---|---|---|---|---|---|---|
| Median | 74.94 | 52.00 | 71.75 | 71.72 | 54.01 | 48.45 | 44.22 | 58.80 | 78.54 | 74.48 | 75.55 |
| Mean | 76.77 | 54.19 | 73.95 | 74.18 | 59.90 | 54.95 | 57.81 | 64.99 | 81.44 | 77.59 | 76.35 |
| Most_freq | 76.85 | 74.91 | 87.86 | 134.91 | 57.42 | 55.29 | 73.15 | 120.96 | 138.22 | 115.75 | 108.77 |
| EM | 54.85 | 34.33 | 49.01 | 10.44 | 56.00 | 28.19 | 40.18 | 37.90 | 69.71 | 69.62 | 63.24 |
| MICE | 80.96 | 65.17 | 75.57 | 17.54 | 97.94 | 59.57 | 76.17 | 61.80 | 90.74 | 83.85 | 74.30 |
| MIRACLE | 70.32 | 58.43 | 43.20 | 10.66 | 59.03 | 32.11 | 78.68 | 38.90 | 334.45 | 357.93 | 335.48 |
| Softimpute | 64.87 | 51.11 | 57.19 | 28.09 | 61.82 | 38.95 | 55.44 | 62.52 | 74.64 | 76.66 | 68.72 |
| MissForest | 58.57 | 36.54 | 50.05 | 25.92 | 65.17 | 33.95 | 45.75 | 39.32 | 69.28 | 55.996 | 63.50 |
| KNN | 61.83 | 36.61 | 54.97 | 12.55 | 69.90 | 33.87 | 46.43 | 41.60 | 91.01 | 71.14 | 69.51 |
| ReMasker | 34.30 | 32.17 | 45.25 | 12.48 | 52.46 | 30.06 | 41.18 | 29.57 | 71.45 | 59.04 | 66.66 |
| HyperImpute | 37.19 | 36.54 | 41.89 | 10.73 | 55.01 | 27.57 | 37.16 | 26.62 | 83.01 | 62.65 | 59.23 |
| GRAPE | 41.25 | 39.46 | 41.96 | 12.88 | 53.93 | 40.15 | 52.43 | 51.57 | 76.94 | 67.12 | 70.10 |
| IGRM | 41.85 | 39.73 | 41.87 | 12.70 | 54.84 | 43.79 | 58.83 | 47.50 | 77.02 | 69.70 | 69.30 |
| MIWAE | 81.48 | 57.33 | 76.58 | 76.64 | 58.61 | 51.31 | 46.29 | 63.23 | 88.11 | 82.94 | 85.54 |
| GAIN | 64.94 | 74.07 | 58.85 | 34.79 | 118.31 | 70.74 | 47.82 | 55.75 | 75.75 | 63.73 | 69.49 |
| MCFlow | 58.06 | 59.43 | 53.71 | 23.87 | 76.74 | 70.51 | 82.18 | 52.61 | 100.74 | 106.67 | 100.23 |
| TabCSDI | 76.87 | 54.67 | 73.86 | 75.41 | 62.78 | 54.75 | 58.18 | 65.01 | 87.57 | 88.46 | 84.24 |
| Missdiff | 70.46 | 139.05 | 137.64 | 56.93 | 58.19 | 204.29 | 117.99 | 57.27 | 154.91 | 172.34 | 103.98 |
| DiffPuter | 30.45 | 32.60 | 41.87 | 12.72 | 50.93 | 23.45 | 35.16 | 28.93 | 77.93 | 68.37 | 75.98 |
| **Impute-MACFM** | 32.87 | 21.23 | 40.16 | 11.84 | 45.88 | 22.63 | 34.19 | 28.73 | 53.95 | 35.38 | 37.22 |

Table 10: Out-of-sample imputation performance under MCAR mechanism. (MAE ↓)

| | **Dataset** | | | | | | | | | | |
|---|---|---|---|---|---|---|---|---|---|---|---|
| **Method** | Letter | Gesture | Magic | Bean | Adult | Default | Shoppers | News | AHEI-S1 | AHEI-S3 | AHEI-S4 |
| Median | 75.23 | 51.22 | 72.51 | 71.63 | 53.44 | 48.28 | 45.65 | 58.13 | 72.84 | 80.23 | 78.45 |
| Mean | 76.90 | 53.28 | 74.61 | 73.61 | 59.35 | 54.82 | 58.54 | 64.28 | 77.09 | 80.70 | 79.12 |
| Most_freq | 77.27 | 74.47 | 88.59 | 135.23 | 56.78 | 54.93 | 75.10 | 120.79 | 145.34 | 120.29 | 111.86 |
| MICE | 81.06 | 64.75 | 75.71 | 16.95 | 98.47 | 59.27 | 76.42 | 66.21 | 91.33 | 89.91 | 78.69 |
| MIRACLE | 62.06 | 56.48 | 44.09 | 13.88 | 60.41 | 57.98 | 65.26 | 63.57 | 332.62 | 320.42 | 302.17 |
| Softimpute | 64.81 | 51.00 | 57.19 | 27.40 | 62.53 | 43.61 | 55.96 | 65.50 | 80.30 | 80.04 | 77.38 |
| MissForest | 58.79 | 35.64 | 50.42 | 25.48 | 64.33 | 33.87 | 45.54 | 39.18 | 67.94 | 60.65 | 68.76 |
| KNN | 62.16 | 39.12 | 53.31 | 12.02 | 70.21 | 31.77 | 49.28 | 48.20 | 91.78 | 75.82 | 97.64 |
| ReMasker | 34.68 | 32.00 | 45.89 | 13.50 | 51.09 | 30.15 | 42.08 | 32.39 | 81.03 | 73.57 | 74.89 |
| HyperImpute | 41.45 | 38.12 | 43.04 | 12.41 | 55.52 | 26.23 | 36.93 | 44.76 | 87.83 | 69.65 | 86.95 |
| GRAPE | 42.15 | 101.48 | 54.50 | 17.83 | 58.07 | 59.35 | 49.45 | 73.33 | 81.10 | 76.92 | 83.03 |
| IGRM | 141.85 | 152.83 | 146.79 | 113.27 | 141.45 | 156.35 | 114.54 | 124.25 | 134.03 | 134.03 | 129.81 |
| MIWAE | 81.88 | 56.53 | 76.83 | 76.53 | 58.26 | 50.96 | 47.81 | 62.51 | 85.68 | 84.11 | 92.85 |
| GAIN | 67.21 | 59.39 | 60.88 | 34.90 | 120.37 | 53.14 | 47.28 | 80.42 | 110.66 | 102.23 | 126.45 |
| MCFlow | 58.40 | 81.71 | 55.93 | 25.13 | 75.88 | 68.71 | 79.46 | 87.69 | 99.85 | 99.64 | 97.84 |
| TabCSDI | 77.13 | 53.91 | 74.40 | 74.83 | 61.88 | 54.45 | 59.34 | 65.36 | 84.98 | 89.85 | 87.19 |
| Missdiff | 70.96 | 129.53 | 141.02 | 53.51 | 58.42 | 184.16 | 85.65 | 62.51 | 262.42 | 216.98 | 119.22 |
| DiffPuter | 29.38 | 28.98 | 42.11 | 12.22 | 49.38 | 23.02 | 36.11 | 28.49 | 77.81 | 68.22 | 74.57 |
| **Impute-MACFM** | 36.34 | 28.10 | 42.95 | 12.25 | 48.44 | 24.56 | 37.52 | 31.45 | 69.89 | 64.05 | 60.59 |

Table 11: In-sample imputation performance under MCAR mechanism. (RMSE ↓)

| | **Dataset** | | | | | | | | | | |
|---|---|---|---|---|---|---|---|---|---|---|---|
| **Method** | Letter | Gesture | Magic | Bean | Adult | Default | Shoppers | News | AHEI-S1 | AHEI-S3 | AHEI-S4 |
| Median | 101.28 | 101.32 | 101.55 | 102.07 | 100.18 | 103.33 | 105.26 | 104.07 | 105.16 | 105.03 | 104.09 |
| Mean | 99.95 | 99.50 | 99.20 | 99.99 | 99.35 | 99.06 | 99.34 | 99.28 | 103.65 | 99.71 | 103.10 |
| Most_freq | 105.38 | 133.79 | 122.50 | 177.60 | 106.15 | 114.51 | 149.14 | 193.38 | 180.25 | 155.88 | 148.42 |
| EM | 74.51 | 68.05 | 74.07 | 28.08 | 92.98 | 82.94 | 75.35 | 63.42 | 92.07 | 101.04 | 91.65 |
| MICE | 105.47 | 93.85 | 104.91 | 41.34 | 131.94 | 106.68 | 109.04 | 89.53 | 118.10 | 112.59 | 107.48 |
| MIRACLE | 103.47 | 107.54 | 70.05 | 30.40 | 103.94 | 87.80 | 146.81 | 77.31 | 364.18 | 396.94 | 369.99 |
| Softimpute | 86.54 | 94.25 | 83.86 | 47.83 | 99.23 | 78.45 | 89.80 | 91.65 | 95.41 | 99.00 | 95.41 |
| MissForest | 80.12 | 74.74 | 76.23 | 44.10 | 117.58 | 79.84 | 89.37 | 69.17 | 94.24 | 83.14 | 90.81 |
| KNN | 83.67 | 71.18 | 83.38 | 31.71 | 107.69 | 74.81 | 83.63 | 70.12 | 121.91 | 107.93 | 100.04 |
| ReMasker | 48.56 | 68.34 | 69.83 | 31.44 | 90.98 | 70.61 | 79.12 | 59.02 | 95.88 | 86.997 | 92.46 |
| HyperImpute | 53.91 | 71.46 | 68.65 | 29.29 | 95.85 | 66.95 | 78.95 | 55.22 | 109.36 | 94.55 | 88.50 |
| GRAPE | 56.74 | 74.13 | 66.83 | 32.11 | 96.06 | 84.01 | 90.38 | 81.10 | 103.18 | 95.47 | 96.01 |
| IGRM | 57.32 | 74.97 | 67.20 | 32.38 | 95.50 | 84.04 | 93.35 | 77.14 | 106.24 | 99.53 | 94.47 |
| MIWAE | 105.30 | 103.21 | 104.80 | 105.81 | 103.36 | 104.26 | 105.49 | 104.87 | 114.58 | 108.77 | 114.34 |
| GAIN | 86.55 | 113.49 | 83.36 | 54.77 | 173.99 | 119.37 | 103.92 | 89.95 | 99.86 | 92.84 | 96.73 |
| MCFlow | 77.96 | 102.54 | 85.68 | 44.04 | 120.78 | 108.39 | 124.22 | 85.40 | 130.91 | 136.99 | 132.88 |
| TabCSDI | 100.13 | 99.71 | 99.34 | 100.39 | 99.79 | 99.09 | 99.40 | 99.84 | 109.44 | 113.81 | 115.63 |
| Missdiff | 92.08 | 178.18 | 311.00 | 142.13 | 98.39 | 346.37 | 210.79 | 175.48 | 368.28 | 517.89 | 199.12 |
| DiffPuter | 47.18 | 76.25 | 68.27 | 33.91 | 96.77 | 80.99 | 78.66 | 57.70 | 104.28 | 93.24 | 106.35 |
| **Impute-MACFM** | 48.30 | 39.45 | 61.29 | 32.21 | 85.73 | 52.48 | 69.55 | 49.79 | 74.35 | 49.45 | 51.95 |

Table 12: Out-of-sample imputation performance under MCAR mechanism. (RMSE ↓)

| Method | Letter | Gesture | Magic | Bean | Adult | Default | Shoppers | News | AHEI-S1 | AHEI-S3 | AHEI-S4 |
|---|---|---|---|---|---|---|---|---|---|---|---|
| Median | 101.31 | 97.70 | 101.68 | 100.77 | 98.42 | 100.96 | 104.39 | 102.33 | 114.08 | 110.72 | 107.01 |
| Mean | 99.93 | 95.85 | 99.26 | 98.50 | 97.59 | 96.68 | 97.88 | 97.57 | 111.73 | 100.83 | 105.80 |
| Most_freq | 105.33 | 130.89 | 122.68 | 177.05 | 104.27 | 112.29 | 149.05 | 192.77 | 190.24 | 153.93 | 149.22 |
| MICE | 105.51 | 92.01 | 104.30 | 39.95 | 130.87 | 100.61 | 109.24 | 666.90 | 121.60 | 113.73 | 110.95 |
| MIRACLE | 93.79 | 112.32 | 70.33 | 31.96 | 101.75 | 122.00 | 119.98 | 429.98 | 366.21 | 346.50 | 341.81 |
| Softimpute | 86.24 | 90.41 | 83.56 | 45.49 | 97.43 | 81.17 | 90.41 | 164.37 | 111.59 | 100.51 | 108.95 |
| MissForest | 80.38 | 70.21 | 76.54 | 41.29 | 115.36 | 78.02 | 89.17 | 68.64 | 101.08 | 86.08 | 99.47 |
| KNN | 84.13 | 72.01 | 79.58 | 29.45 | 106.07 | 68.52 | 88.70 | 78.44 | 122.69 | 101.03 | 139.12 |
| ReMasker | 49.11 | 64.32 | 70.72 | 31.66 | 88.41 | 69.42 | 79.46 | 63.41 | 116.56 | 100.42 | 101.55 |
| HyperImpute | 59.36 | 70.73 | 70.18 | 28.86 | 94.65 | 61.26 | 77.68 | 78.58 | 117.76 | 95.61 | 121.60 |
| GRAPE | 57.96 | 134.41 | 81.79 | 36.13 | 94.73 | 96.01 | 87.88 | 112.66 | 117.01 | 103.07 | 110.79 |
| IGRM | 167.90 | 176.94 | 174.95 | 140.98 | 172.28 | 192.02 | 157.62 | 162.10 | 167.79 | 167.79 | 159.82 |
| MIWAE | 105.86 | 99.46 | 104.50 | 104.80 | 102.01 | 101.69 | 104.52 | 103.21 | 123.31 | 108.74 | 125.42 |
| GAIN | 89.05 | 94.52 | 84.92 | 53.99 | 178.77 | 96.54 | 100.00 | 142.19 | 143.90 | 136.66 | 168.23 |
| MCFlow | 78.54 | 116.66 | 90.28 | 43.96 | 117.43 | 108.70 | 124.28 | 121.94 | 139.23 | 128.29 | 127.27 |
| TabCSDI | 100.19 | 96.05 | 99.40 | 99.01 | 98.23 | 96.99 | 97.94 | 98.37 | 118.55 | 119.40 | 118.83 |
| Missdiff | 92.42 | 179.02 | 317.13 | 116.06 | 99.02 | 447.43 | 197.29 | 192.19 | 450.01 | 598.06 | 196.98 |
| DiffPuter | 45.90 | 65.30 | 68.82 | 31.57 | 92.04 | 71.78 | 77.98 | 56.16 | 113.47 | 92.98 | 101.56 |
| **Impute-MACFM** | 52.81 | 58.24 | 68.01 | 32.05 | 91.73 | 61.87 | 74.50 | 55.78 | 94.16 | 85.06 | 84.06 |

Table 13: In-sample imputation performance under MNAR mechanism. (MAE ↓)

| Method | Letter | Gesture | Magic | Bean | Adult | Default | Shoppers | News | AHEI-S1 | AHEI-S3 | AHEI-S4 |
|---|---|---|---|---|---|---|---|---|---|---|---|
| Median | 76.15 | 55.94 | 76.83 | 71.44 | 54.18 | 49.96 | 50.49 | 58.25 | 75.63 | 79.20 | 74.61 |
| Mean | 78.44 | 58.25 | 77.93 | 76.33 | 60.20 | 56.92 | 61.91 | 65.13 | 79.36 | 79.79 | 75.64 |
| Most_freq | 77.81 | 92.95 | 111.56 | 127.74 | 57.29 | 56.50 | 76.46 | 120.49 | 147.50 | 116.88 | 105.22 |
| EM | 55.99 | 37.63 | 50.87 | 10.26 | – | 28.69 | – | – | 68.68 | 69.61 | 63.67 |
| MICE | 81.27 | 66.38 | 75.42 | 16.80 | 97.86 | 58.54 | 76.85 | 62.79 | 94.08 | 81.57 | 75.76 |
| MIRACLE | 70.57 | 67.05 | 47.57 | 11.90 | 62.28 | 30.68 | 73.85 | 40.06 | 347.51 | 334.85 | 324.13 |
| Softimpute | 65.15 | 56.97 | 61.80 | 27.86 | 61.99 | 43.88 | 58.99 | 62.19 | 78.40 | 69.39 | 68.40 |
| MissForest | 59.08 | 40.62 | 50.88 | 24.67 | 62.92 | 33.84 | 51.68 | 38.03 | 68.60 | 62.80 | 63.86 |
| KNN | 62.70 | 40.68 | 59.97 | 12.50 | 72.46 | 34.20 | 46.52 | 42.36 | 86.10 | 70.84 | 73.83 |
| ReMasker | 35.29 | 35.51 | 74.77 | 10.82 | 50.03 | 31.11 | 37.86 | 30.54 | 73.12 | 64.86 | 66.69 |
| HyperImpute | 44.89 | 36.73 | 44.04 | 12.74 | 54.35 | 25.09 | 38.12 | – | 78.02 | 64.75 | 67.56 |
| GRAPE | 42.65 | 38.81 | 41.27 | 11.57 | 57.27 | 39.09 | 54.83 | 47.50 | 77.55 | 71.27 | 71.48 |
| IGRM | 40.86 | 46.47 | 40.56 | 11.84 | 54.85 | 45.68 | 61.39 | 44.99 | 79.83 | 70.24 | 71.12 |
| MIWAE | 82.56 | 60.83 | 81.61 | 76.39 | 59.06 | 52.97 | 51.48 | 62.80 | 85.55 | 86.96 | 86.00 |
| GAIN | 68.94 | 76.43 | 64.67 | 36.64 | 95.82 | 77.97 | 52.02 | 54.27 | 82.31 | 81.91 | 73.66 |
| MCFlow | 59.55 | 65.37 | 59.05 | 24.57 | 79.03 | 72.51 | 79.61 | 52.28 | 100.51 | 97.95 | 95.17 |
| TabCSDI | 77.69 | 58.21 | 78.19 | 72.27 | 63.10 | 55.53 | 61.82 | 64.82 | 84.21 | 87.76 | 84.65 |
| Missdiff | 67.91 | 434.02 | 71.79 | 54.90 | 59.75 | 482.14 | 98.13 | 196.43 | 226.72 | 205.66 | 149.40 |
| DiffPuter | 35.03 | 41.84 | 46.82 | 13.71 | 50.91 | 28.92 | 47.98 | 30.92 | 75.29 | 71.39 | 70.34 |
| **Impute-MACFM** | 24.59 | 24.18 | 40.16 | 11.33 | 37.32 | 19.21 | 25.26 | 23.23 | 51.65 | 57.60 | 59.01 |

Table 14: In-sample imputation performance under MNAR mechanism. (RMSE ↓)

| Method | Letter | Gesture | Magic | Bean | Adult | Default | Shoppers | News | AHEI-S1 | AHEI-S3 | AHEI-S4 |
|---|---|---|---|---|---|---|---|---|---|---|---|
| Median | 102.43 | 108.72 | 109.32 | 98.85 | 101.90 | 109.37 | 116.83 | 103.40 | 103.39 | 110.34 | 102.62 |
| Mean | 101.31 | 107.18 | 106.36 | 99.24 | 101.10 | 106.16 | 108.37 | 99.24 | 102.33 | 102.45 | 101.88 |
| Most_freq | 106.08 | 161.45 | 152.32 | 170.39 | 107.09 | 119.99 | 153.24 | 192.66 | 186.04 | 157.25 | 143.69 |
| EM | 77.17 | 74.80 | 77.74 | 26.91 | – | 74.18 | – | – | 95.85 | 96.72 | 95.22 |
| MICE | 106.63 | 97.61 | 105.68 | 40.13 | 133.29 | 97.70 | 111.02 | 90.02 | 125.42 | 110.13 | 106.07 |
| MIRACLE | 108.02 | 119.41 | 79.83 | 36.91 | 114.09 | 86.12 | 139.96 | 78.44 | 384.18 | 368.72 | 358.27 |
| Softimpute | 87.56 | 105.55 | 93.52 | 45.63 | 101.09 | 87.22 | 96.29 | 91.48 | 101.49 | 93.64 | 95.14 |
| MissForest | 81.29 | 82.31 | 77.41 | 39.22 | 109.56 | 86.64 | 104.78 | 66.88 | 96.90 | 92.72 | 91.05 |
| KNN | 86.29 | 78.63 | 90.50 | 29.83 | 111.05 | 80.74 | 86.95 | 71.18 | 118.35 | 111.09 | 105.29 |
| ReMasker | 50.92 | 76.09 | 102.47 | 26.60 | 89.42 | 76.46 | 77.31 | 61.08 | 99.49 | 94.83 | 90.80 |
| HyperImpute | 63.15 | 72.66 | 73.07 | 30.30 | 96.90 | 76.89 | 83.32 | – | 108.59 | 98.07 | 98.71 |
| GRAPE | 58.75 | 77.33 | 65.97 | 28.11 | 95.70 | 88.17 | 96.89 | 78.17 | 107.74 | 103.05 | 99.19 |
| IGRM | 56.57 | 90.49 | 66.08 | 28.24 | 96.24 | 90.36 | 96.71 | 75.96 | 110.08 | 100.26 | 98.64 |
| MIWAE | 106.31 | 110.22 | 112.68 | 102.73 | 105.12 | 110.37 | 114.96 | 104.18 | 114.09 | 112.92 | 113.65 |
| GAIN | 92.03 | 118.86 | 89.96 | 55.19 | 146.82 | 136.27 | 113.91 | 88.67 | 111.37 | 119.73 | 104.52 |
| MCFlow | 81.02 | 117.44 | 89.89 | 41.85 | 121.88 | 114.29 | 120.99 | 83.68 | 131.30 | 126.04 | 124.34 |
| TabCSDI | 100.73 | 106.90 | 106.14 | 95.81 | 101.77 | 104.64 | 106.75 | 99.52 | 107.15 | 114.38 | 115.96 |
| Missdiff | 112.76 | 2692.45 | 279.40 | 92.32 | 108.83 | 14797.81 | 311.02 | 7985.34 | 865.42 | 473.04 | 764.29 |
| DiffPuter | 53.94 | 99.50 | 78.29 | 44.65 | 95.65 | 99.41 | 126.34 | 58.16 | 101.20 | 99.38 | 101.62 |
| **Impute-MACFM** | 38.75 | 40.78 | 60.42 | 32.99 | 70.44 | 45.14 | 55.73 | 41.35 | 69.43 | 80.95 | 80.27 |

Table 15: Out-of-sample imputation performance under MNAR mechanism. (RMSE ↓)

| Method | Letter | Gesture | Magic | Bean | Adult | Default | Shoppers | News | AHEI-S1 | AHEI-S3 | AHEI-S4 |
|---|---|---|---|---|---|---|---|---|---|---|---|
| Median | 102.08 | 100.81 | 95.84 | 115.95 | 100.86 | 95.70 | 92.88 | 1967.81 | 125.11 | 102.89 | 115.33 |
| Mean | 100.26 | 98.81 | 94.46 | 111.31 | 99.82 | 94.31 | 87.88 | 1967.61 | 122.50 | 94.11 | 114.33 |
| Most_freq | 105.90 | 154.63 | 143.79 | 185.72 | 105.88 | 106.45 | 144.34 | 1975.23 | 191.78 | 158.68 | 142.55 |
| MICE | 105.76 | 91.49 | 98.51 | 52.68 | 132.75 | 93.86 | 103.72 | 1688.51 | 123.82 | 110.14 | 110.17 |
| MIRACLE | 119.18 | 117.19 | 64.05 | 36.83 | 129.31 | 94.78 | 116.51 | 1962.45 | 397.62 | 361.50 | 403.26 |
| Softimpute | 87.38 | 93.47 | 78.02 | 49.53 | 108.39 | 86.69 | 84.97 | 1866.97 | 122.41 | 92.55 | 111.59 |
| MissForest | 79.84 | 72.32 | 70.31 | 62.53 | 103.81 | 87.91 | 93.40 | 1966.28 | 95.31 | 77.76 | 99.87 |
| KNN | 84.47 | 71.70 | 78.24 | 29.73 | 109.90 | 74.03 | 80.41 | 2021.41 | 138.43 | 103.38 | 129.23 |
| ReMasker | 50.38 | 70.57 | 94.17 | 29.01 | 91.75 | 73.94 | 69.94 | 1966.88 | 119.77 | 90.74 | 110.31 |
| HyperImpute | 68.24 | 72.15 | 65.96 | 62.60 | 98.08 | 80.27 | 75.00 | 83.97 | 118.90 | 96.23 | 122.78 |
| GRAPE | 59.72 | 127.84 | 75.99 | 34.63 | 97.15 | 97.13 | 82.20 | 196.64 | 121.65 | 99.72 | 115.02 |
| IGRM | 164.09 | 176.41 | 143.37 | 190.43 | 144.70 | 163.25 | 158.58 | 197.10 | 161.74 | 132.39 | 146.35 |
| MIWAE | 105.71 | 102.42 | 99.52 | 118.93 | 103.56 | 96.95 | 92.39 | 1967.89 | 134.87 | 104.95 | 126.31 |
| GAIN | 90.15 | 95.37 | 77.47 | 68.14 | 176.37 | 102.35 | 90.73 | 1969.52 | 144.43 | 171.99 | 171.06 |
| MCFlow | 80.07 | 127.18 | 81.43 | 63.73 | 127.64 | 116.38 | 104.93 | 196.94 | 143.98 | 124.73 | 128.52 |
| TabCSDI | 100.52 | 99.09 | 94.19 | 112.46 | 99.67 | 94.24 | 87.71 | 1967.72 | 131.09 | 112.30 | 127.57 |
| Missdiff | 96.07 | 3509.05 | 155.11 | 101.52 | 106.04 | 9941.52 | 407.98 | 5667.95 | 957.41 | 480.39 | 219.87 |
| DiffPuter | 48.85 | 77.82 | 68.86 | 35.98 | 93.26 | 88.29 | 86.36 | 59.47 | 104.74 | 90.05 | 105.57 |
| **Impute-MACFM** | 50.61 | 66.44 | 74.60 | 41.46 | 88.19 | 68.07 | 76.42 | 53.51 | 95.94 | 92.55 | 94.18 |

Table 16: **Downstream macro AUROC on five UCI datasets.** We report mean AUROC across 5 independent runs. The best results are highlighted in **bold**.

| Method | ADULT | DEFAULT | MAGIC | SHOPPERS | BEAN |
|---|---|---|---|---|---|
| GAIN | 0.7968±0.004 | 0.7155±0.003 | 0.8002±0.003 | 0.8247±0.003 | 0.9859±0.002 |
| HyperImpute | 0.8726±0.002 | 0.7261±0.003 | 0.8184±0.002 | 0.8330±0.002 | 0.9897±0.001 |
| ICE | 0.8517±0.003 | 0.7203±0.002 | 0.7852±0.004 | 0.8160±0.003 | **0.9944±0.001** |
| MICE | 0.8297±0.003 | 0.7120±0.004 | 0.7679±0.004 | 0.7930±0.003 | 0.9848±0.002 |
| MIRACLE | 0.8626±0.002 | 0.7207±0.003 | 0.8101±0.003 | 0.8239±0.002 | 0.9925±0.001 |
| MIWAE | 0.8458±0.003 | 0.7140±0.003 | 0.7871±0.003 | 0.8282±0.003 | 0.9637±0.003 |
| Most-frequent | 0.8424±0.003 | 0.7188±0.002 | 0.7375±0.001 | 0.8272±0.002 | 0.9622±0.001 |
| MissForest | 0.8648±0.002 | 0.7206±0.002 | 0.7899±0.003 | 0.8427±0.002 | 0.9909±0.001 |
| SoftImpute | 0.8325±0.003 | 0.7144±0.003 | 0.7865±0.003 | 0.8339±0.002 | 0.9925±0.002 |
| MCFlow | 0.8156±0.004 | 0.7019±0.004 | 0.7867±0.004 | 0.6907±0.005 | 0.9691±0.003 |
| Remasker | 0.8713±0.002 | 0.7333±0.003 | 0.7902±0.003 | 0.8443±0.002 | 0.9745±0.002 |
| GRAPE | 0.8766±0.002 | 0.7177±0.003 | 0.8017±0.003 | 0.8274±0.002 | 0.9830±0.001 |
| DiffPuter | 0.8552±0.003 | 0.7208±0.003 | 0.8151±0.002 | 0.8319±0.002 | 0.9921±0.001 |
| **Impute-MACFM** | **0.8818±0.002** | **0.7480±0.002** | **0.8445±0.003** | **0.8867±0.001** | 0.9918±0.002 |

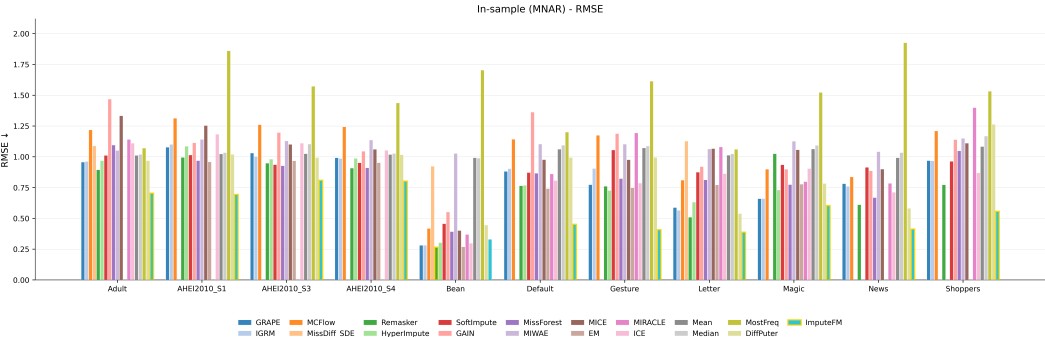

Figure 5: MNAR, In-sample imputation performance on RMSE score (lower is better)

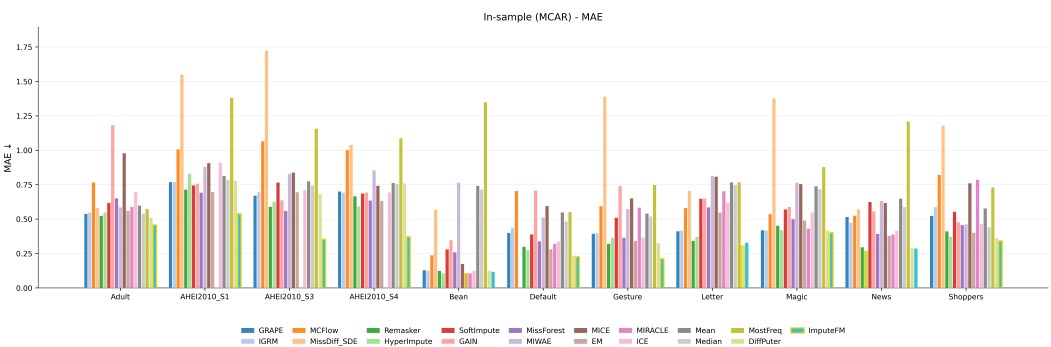

Figure 6: MCAR, In-sample imputation performance on MAE score

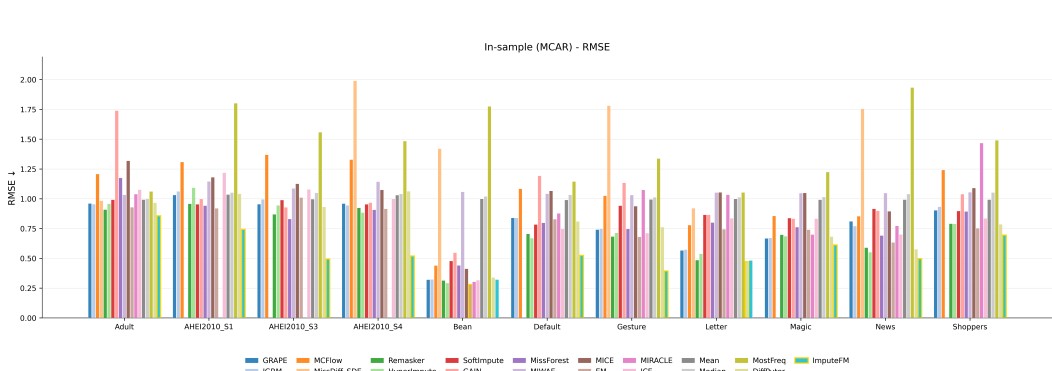

Figure 7: MCAR, In-sample imputation performance on RMSE score

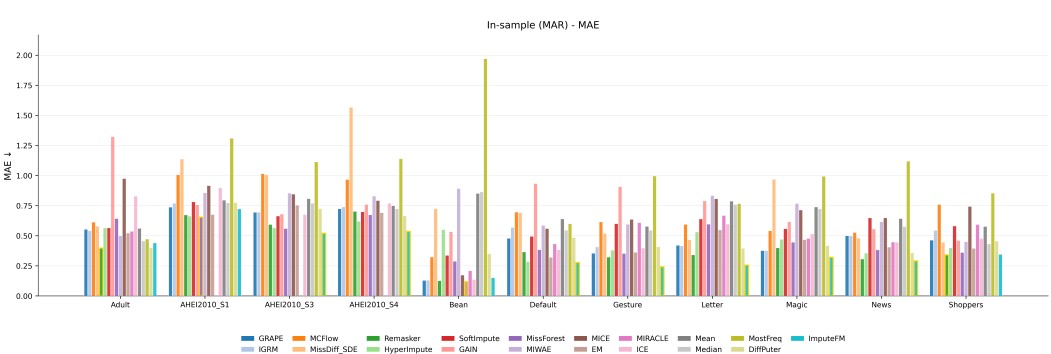

Figure 8: MAR, In-sample imputation performance on MAE score

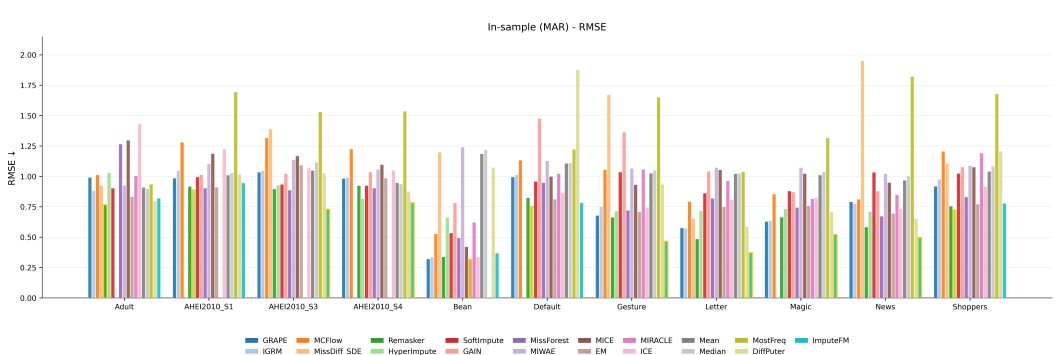

Figure 9: MAR, In-sample imputation performance on RMSE score

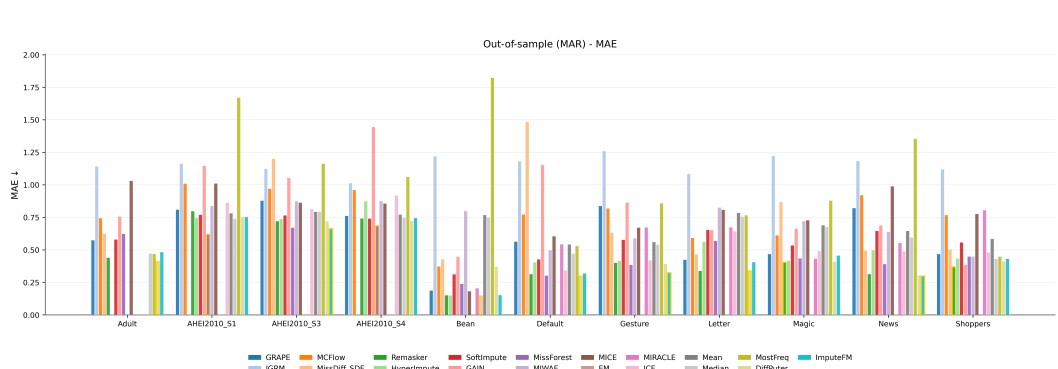

Figure 10: MAR, Out-of-sample imputation performance on MAE score

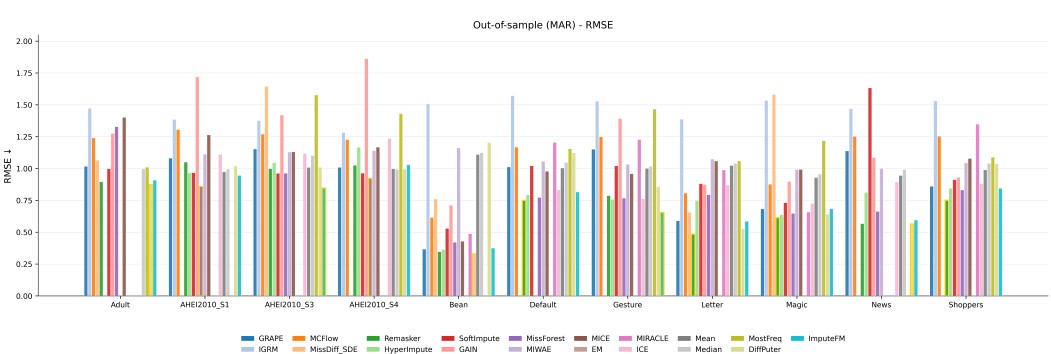

Figure 11: MAR, Out-of-sample imputation performance on RMSE score

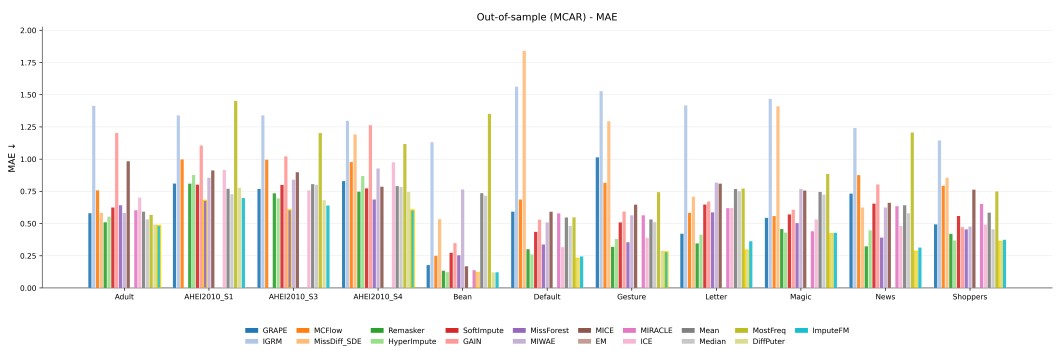

Figure 12: MCAR, Out-of-sample imputation performance on MAE score

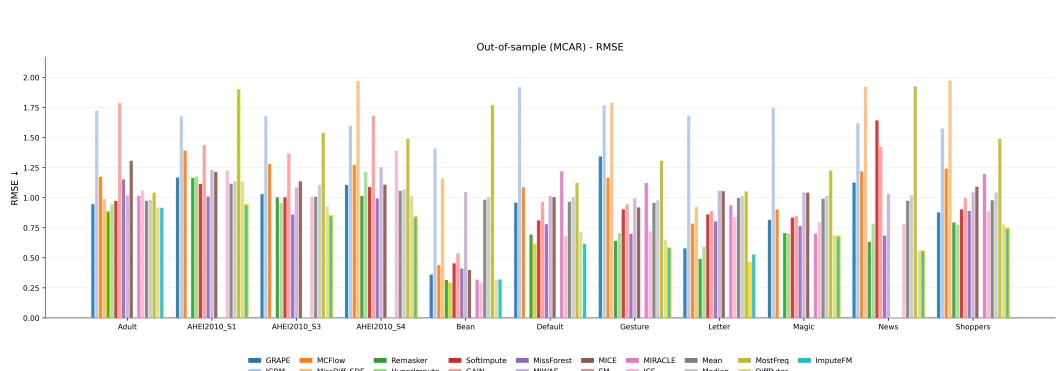

Figure 13: MCAR, Out-of-sample imputation performance on RMSE score

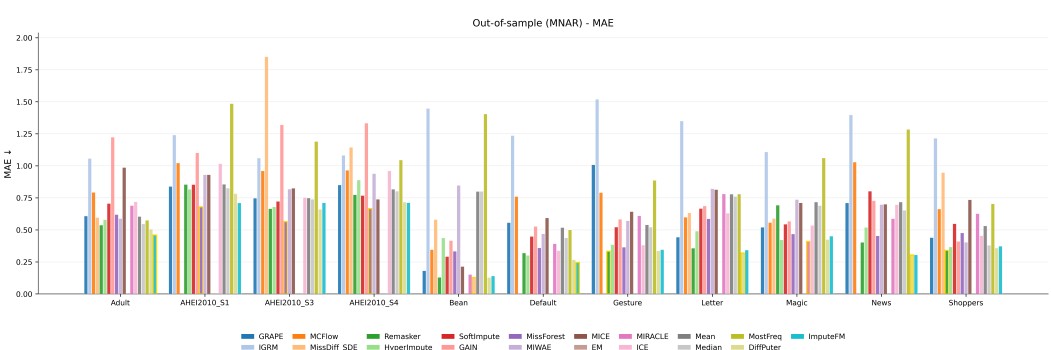

Figure 14: MNAR, Out-of-sample imputation performance on MAE score

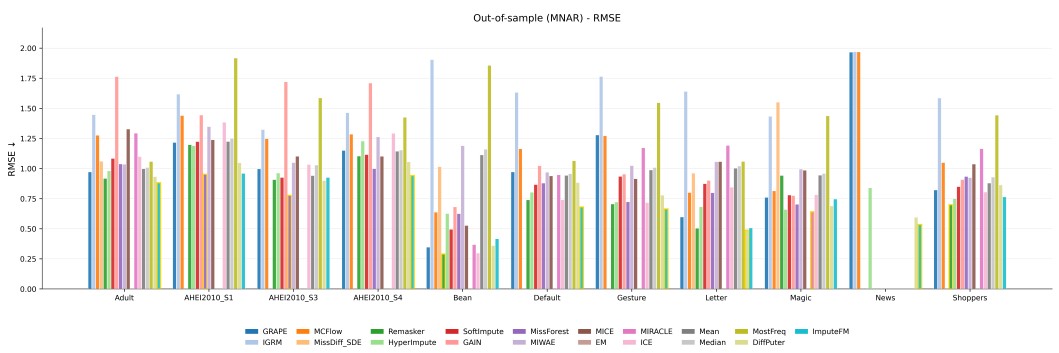

Figure 15: MNAR, Out-of-sample imputation performance on RMSE score

# E USE OF LARGE LANGUAGE MODELS

This paper's core idea, problem formulation, model design, experiments, and writing structure are authored by the paper's authors. Large Language Models (LLMs) were used in a limited assistant role only. Specifically:

**Code assistance:** debugging small issues in our own code (e.g., Python syntax errors, minor API usage), and suggesting non research critical boilerplate (e.g., plotting, simple Bash one-liners). All final implementations were written and verified by the authors.

**Editing support:** light grammar/spell checking and occasional phrasing polish for a small number of sentences to improve readability. Substantive technical content was written by the authors.

**Typesetting:** minor LaTeX formatting suggestions (e.g., table and figure layout) without altering scientific content.

LLMs *were not* used to generate research ideas, design experiments, write the technical sections, analyze results, create or curate datasets, or run/combine results. All methodological decisions, experiment configurations, and conclusions are the authors' own, and every LLM-suggested change was reviewed for accuracy and appropriateness.

