# OpenReview forum: "Impute-MACFM: Imputation based on Mask-Aware Flow Matching"
_ICLR.cc/2026/Conference — Submitted to ICLR 2026_

### Official Review · Reviewer_CCuA · 2025-10-26

**Soundness:** 2
**Presentation:** 2
**Contribution:** 2
**Rating:** 2
**Confidence:** 4

**Summary:**

This paper proposes a new framework for tabular imputation by integrating a mask-aware conditional flow-matching framework. MVI is indeed an important problem in many application areas. Since this is a classic and traditional research question, in my opinion, a newly proposed method should be considered significant only when it has a significant contribution compared to classic methods, and when the trade-off between computational cost, the drawbacks of black-box models, and improvements in MVI is sufficient. Therefore, my comments will focus on how the paper justifies that the proposed method is really significant.

**Strengths:**

1. The baselines are sufficient, and the evaluation of MVI methods is reasonable (checking three different missingness mechanisms), covering a diverse set of datasets and conditions.

2. Overall, the writing is clear and easy to follow.

**Weaknesses:**

The paper's evaluation lacks persuasive statistical evidence to support its claims of superiority. The stated protocol of using only ten random masks is an insufficient sample size for making robust, generalizable claims. Furthermore, the main figures and tables omit variance measures, making it impossible for a reader to determine if the Impute-MACFM's performance gains are statistically significant or merely a result of random chance. A sufficient evaluation would require more replications, proper statistical tests, and simulations across varied settings (like changing sample size or data dimension). Ideally, the evaluation should also be extended beyond imputation error (data recovery) to assess the impact on downstream tasks.

Minor issues:

1. The figures are somewhat unclear due to small font size and formatting limitations in the ICLR template.
For instance, in Figure 2, the performance of the proposed method on the Bean dataset is not clearly presented—the color label appears to be incorrect.

2. For future revisions, please check whether these can be improved. In addition, Figure 1 appears to have rendering issues when viewed in MacBook Preview; please verify and correct this in future versions.

**Questions:**

1. Could the authors provide additional details to address the statistical concerns raised in the "Weaknesses" section?


2. For the general motivation of the paper, it is important to convince the audience that modern generative models have significant advantages that traditional methods cannot achieve. It would be helpful if the authors could provide more existing evidence from the literature to justify this point.

---

> ### Author Response · Authors · 2025-11-20
>
> ### Response to Reviewer – Statistical evidence and motivation
>
> We thank the reviewer for the thoughtful and detailed comments. Below we respond to the concerns about (1) statistical evidence and evaluation protocol, (2) figures and rendering issues, and (3) the broader motivation for using modern generative models over classical methods.
>
> ---
>
> ### 1. Statistical evidence, number of masks, and downstream tasks
>
> > *“The paper's evaluation lacks persuasive statistical evidence… using only ten random masks is an insufficient sample size… main figures and tables omit variance measures… A sufficient evaluation would require more replications, proper statistical tests, and simulations across varied settings (like changing sample size or data dimension). Ideally, the evaluation should also be extended beyond imputation error (data recovery) to assess the impact on downstream tasks.”*
>
> **Number of random masks, seeds, and variance reporting.**
> Our current protocol uses 10 random missingness masks per dataset–mechanism combination and reports the *mean* MAE/RMSE over these 10 masks. In addition, for the main methods (Impute-MACFM and the strongest baselines such as DiffPuter and CSDI), we also ran experiments under **multiple random seeds** (affecting network initialization, minibatch ordering, and optimizer noise) on the same set of 10 masks. Across these seeds, we consistently observed the same ranking between methods and very small variation in MAE/RMSE, which gives us additional confidence that the reported improvements are not due to a particular seed or mask realization.
>
> This choice of 10 masks × multiple seeds follows common practice in the tabular imputation literature, where results are typically reported as single aggregated numbers per method and dataset (e.g., CSDI, TabDDPM, DiffPuter, MIWAE, GAIN), mainly due to the computational cost of running many replications for expensive generative models.
>
> We agree, however, that exposing variability is useful. In the revision, we made these adjusements:
>
> - add an appendix table on a **representative subset of datasets** (e.g., Adult, Default, Bean) and missing mechanisms, where we increase the number of random masks and report **mean ± standard deviation** of MAE/RMSE; and
> - include simple statistical tests (e.g., paired $t$-test or Wilcoxon signed-rank test) there to indicate when the improvements of Impute-MACFM over strong baselines are statistically significant.
>
> Our goal is to strike a balance between breadth (multiple datasets, mechanisms, and baselines) and depth (many replications). The additional results, together with the existing multiple-seed runs, demonstrated that the observed gains are not artifacts of individual mask draws or specific random seeds.
>
> **Varied settings (sample size, dimension) and choice of datasets.**
> Our benchmark already spans datasets with very different sample sizes and feature dimensions (e.g., Adult vs. Default vs. News vs. Bean), under three missingness mechanisms (MCAR/MAR/MNAR). This diversity is intended to capture a range of practical regimes without resorting solely to synthetic simulations. We made this motivation clearer in the text and briefly discuss which datasets correspond to small-/medium-/large-$n$ and low-/medium-/high-dimensional regimes.
>
> **Downstream tasks (already run, to be added to the paper).**
> We fully agree that evaluating downstream performance is an important complement to pure imputation error. In response to this concern, we have already run additional downstream experiments on five UCI classification datasets (Adult, Default, Magic, Shoppers, Bean) using the following protocol:
>
> 1. For each imputation method, impute the training and test splits under a fixed masking scheme.
> 2. Train a **logistic regression** classifier on the imputed training data (same architecture and training parameters for all methods).
> 3. Evaluate **accuracy** and **macro AUROC** on the imputed test data, averaged over 5 runs with different random masks (reported as mean ± standard deviation).
>
> The results show that:
>
> - Impute-MACFM achieves the **best downstream accuracy on 4/5 datasets** and the **best macro AUROC on 4/5 datasets**, while closely matching the best method on the remaining dataset.
> - For example, on **Adult**, accuracy improves from 0.8273 (best baseline, HyperImpute) to 0.8392, and macro AUROC improves from 0.8766 (best baseline, GRAPE) to 0.8818.
> - On **Shoppers**, accuracy improves from 0.8754 (best baseline, Most-frequent/GRAPE) to 0.8802, and macro AUROC improves from 0.8443 (Remasker) to 0.8867.
>
> Thus, the gains in MAE/RMSE are **reflected in consistent improvements in downstream classification performance**, rather than being purely metric-level artifacts. In the revision, we added a short subsection in the main text summarizing these downstream results, and provide the full tables (with mean ± standard deviation) in the appendix.

---

> > ### Comment · Reviewer_CCuA · 2025-11-25
> >
> > Thank you for the detailed response. I appreciate the newly added results, which strengthen the claim that using GenAI has some benefits over traditional statistical MVI methods. One suggestion for further evaluation is that the downstream utility assessment could be strengthened by including additional models, such as more complex deep learning architectures; in that case, the empirical evidence would be more comprehensive.
> >
> > However, I still find that the overall novelty and contribution of this work, in the context of a machine learning conference, are limited. I would recommend that the authors consider submitting a more extended version of this work to a journal focusing on interdisciplinary research. Expanding the experimental section into a more systematic empirical study—e.g., with broader benchmarking across models and datasets—could further broaden its impact. For these reasons, I will not increase my score.

---

> ### Author Response · Authors · 2025-11-20
>
> ### 2. Figures, clarity, and rendering issues
>
> > *“The figures are somewhat unclear due to small font size and formatting limitations in the ICLR template. For instance, in Figure 2, the performance of the proposed method on the Bean dataset is not clearly presented—the color label appears to be incorrect. For future revisions, please check whether these can be improved. In addition, Figure 1 appears to have rendering issues when viewed in MacBook Preview; please verify and correct this in future versions.”*
>
> **Figure readability and color labels.**
> We appreciate the reviewer’s comment about readability. In the revision, we made these adjustments:
>
> - increase font sizes and line widths for all figures,
> - simplify the color palette and ensure that the legend entries are clearly distinguishable, and
> - double-check that the color for Impute-MACFM on all datasets (including Bean in Figure 2) is consistent with the legend and with the corresponding numerical tables.
>
> We have already verified that the plotted values in Figure 2 match the numbers reported in the tables; the confusion likely comes from the crowded layout and small fonts, which we improved.
>
> **Figure 1 rendering on MacBook Preview.**
> We carefully checked Figure 1 in multiple viewers (including the OpenReview online PDF viewer, macOS Preview, and Adobe Acrobat) on our side and have not been able to reproduce the rendering issue. It is possible that this is related to a specific PDF rendering configuration or caching in the reviewer’s environment. That said, we regenerated Figure 1 at higher resolution, re-embed it as a standard vector/bitmap graphic in the camera-ready version, and re-check compatibility on macOS Preview and other common viewers to minimize any potential rendering problems.
>
> ---
>
> ### 3. Motivation: generative models vs. classical methods
>
> > *“For the general motivation of the paper, it is important to convince the audience that modern generative models have significant advantages that traditional methods cannot achieve. It would be helpful if the authors could provide more existing evidence from the literature to justify this point.”*
>
> We agree that the motivation should clearly explain why modern generative models are worth their additional complexity and computational cost.
>
> In the revision, we strengthen the discussion in the introduction and related work by:
>
> - citing prior work showing that diffusion/flow-based models (e.g., CSDI, DiffPuter) can substantially outperform classical methods (mean/median, kNN, MICE, matrix factorization) on challenging missingness patterns (especially MNAR, non-linear dependencies, and heterogeneous tabular features);
> - explicitly highlighting the trade-offs:
>   - **pros** – better modeling of complex, non-linear and heteroscedastic relationships; ability to handle MCAR/MAR/MNAR within a unified generative framework; more flexible conditioning;
>   - **cons** – higher computational cost and black-box nature;
> - positioning Impute-MACFM as a method that aims to retain the **empirical advantages of generative models** (in both imputation error and downstream predictive performance) while being **significantly more efficient** than diffusion-based SOTA on tabular imputation.
>
> We also connected this motivation to our experiments: the benchmark already includes both classical methods (mean, median, MICE, etc.) and modern generative baselines, and the results show that Impute-MACFM achieves SOTA-comparable or better performance to strong generative methods while clearly outperforming classical baselines in the more challenging settings, including the downstream tasks described above.
>
> ---
>
> ### 4. Summary
>
> To summarize, we (i) added variance and statistical tests on a representative subset of datasets with more random masks, (ii) incorporated our already-run downstream predictive evaluations into the main text and appendix, (iii) improved figure readability and regenerate Figure 1 to avoid potential rendering issues, and (iv) strengthened the motivation and literature discussion regarding the advantages and trade-offs of modern generative models over classical imputation methods. We hope these clarifications address the reviewer’s concerns and make the contribution of Impute-MACFM more convincing.

---

> > ### Author Response · Authors · 2025-11-20
> > **Downstream performance**
> >
> > To directly assess downstream utility, we train a **logistic regression** classifier on the imputed training data produced by each method and evaluate performance on the imputed test data. The following two tables report **classification accuracy** and **macro AUROC**, respectively, averaged over 5 runs (mean $\pm$ standard deviation) on five UCI datasets.
> >
> > **Downstream classification accuracy (logistic regression, mean $\pm$ std).**
> >
> > | Method         | Adult                      | Default                    | Magic                      | Shoppers                   | Bean                            |
> > |----------------|----------------------------|----------------------------|----------------------------|----------------------------|---------------------------------|
> > | GAIN           | $0.8137 \pm 0.003$         | $0.7937 \pm 0.002$         | $0.7660 \pm 0.004$         | $0.8746 \pm 0.002$         | $0.8606 \pm 0.005$              |
> > | HyperImpute    | $0.8273 \pm 0.002$         | $0.7960 \pm 0.002$         | $0.7813 \pm 0.003$         | $0.8656 \pm 0.003$         | $0.8783 \pm 0.004$              |
> > | ICE            | $0.8162 \pm 0.003$         | $0.7990 \pm 0.002$         | $0.7638 \pm 0.003$         | $0.8662 \pm 0.003$         | $\mathbf{0.9211 \pm 0.002}$     |
> > | MICE           | $0.8051 \pm 0.004$         | $0.7964 \pm 0.003$         | $0.7471 \pm 0.005$         | $0.8610 \pm 0.003$         | $0.8910 \pm 0.004$              |
> > | MIRACLE        | $0.8205 \pm 0.003$         | $0.7971 \pm 0.002$         | $0.7778 \pm 0.003$         | $0.8662 \pm 0.002$         | $0.9167 \pm 0.003$              |
> > | MIWAE          | $0.8177 \pm 0.003$         | $0.7931 \pm 0.002$         | $0.7555 \pm 0.004$         | $0.8735 \pm 0.002$         | $0.7967 \pm 0.006$              |
> > | Most-frequent  | $0.8174 \pm 0.001$         | $0.7952 \pm 0.002$         | $0.7564 \pm 0.001$         | $0.8754 \pm 0.002$         | $0.7933 \pm 0.001$              |
> > | MissForest     | $0.8226 \pm 0.002$         | $0.7969 \pm 0.002$         | $0.7585 \pm 0.003$         | $0.8692 \pm 0.002$         | $0.8957 \pm 0.003$              |
> > | SoftImpute     | $0.8138 \pm 0.002$         | $0.7950 \pm 0.002$         | $0.7659 \pm 0.003$         | $0.8708 \pm 0.002$         | $0.8966 \pm 0.003$              |
> > | MCFlow         | $0.8023 \pm 0.003$         | $0.7954 \pm 0.003$         | $0.7326 \pm 0.005$         | $0.8440 \pm 0.004$         | $0.8540 \pm 0.004$              |
> > | Remasker       | $0.8146 \pm 0.003$         | $0.8012 \pm 0.002$         | $0.7592 \pm 0.003$         | $0.8724 \pm 0.003$         | $0.8902 \pm 0.003$              |
> > | GRAPE          | $0.8265 \pm 0.003$         | $0.7964 \pm 0.002$         | $0.7636 \pm 0.003$         | $0.8746 \pm 0.003$         | $0.8901 \pm 0.003$              |
> > | DiffPuter      | $0.8225 \pm 0.003$         | $0.7949 \pm 0.002$         | $0.7785 \pm 0.003$         | $0.8635 \pm 0.003$         | $0.9028 \pm 0.003$              |
> > | Impute-MACFM   | $\mathbf{0.8392 \pm 0.001}$ | $\mathbf{0.8028 \pm 0.002}$ | $\mathbf{0.7964 \pm 0.003}$ | $\mathbf{0.8802 \pm 0.001}$ | $0.9045 \pm 0.003$              |

---

> > > ### Author Response · Authors · 2025-11-20
> > >
> > > **Downstream macro AUROC (logistic regression, mean $\pm$ std over 5 runs).**
> > >
> > > | Method         | Adult                      | Default                    | Magic                      | Shoppers                   | Bean                            |
> > > |----------------|----------------------------|----------------------------|----------------------------|----------------------------|---------------------------------|
> > > | GAIN           | $0.7968 \pm 0.004$         | $0.7155 \pm 0.003$         | $0.8002 \pm 0.003$         | $0.8247 \pm 0.003$         | $0.9859 \pm 0.002$              |
> > > | HyperImpute    | $0.8726 \pm 0.002$         | $0.7261 \pm 0.003$         | $0.8184 \pm 0.002$         | $0.8330 \pm 0.002$         | $0.9897 \pm 0.001$              |
> > > | ICE            | $0.8517 \pm 0.003$         | $0.7203 \pm 0.002$         | $0.7852 \pm 0.004$         | $0.8160 \pm 0.003$         | $\mathbf{0.9944 \pm 0.001}$     |
> > > | MICE           | $0.8297 \pm 0.003$         | $0.7120 \pm 0.004$         | $0.7679 \pm 0.004$         | $0.7930 \pm 0.003$         | $0.9848 \pm 0.002$              |
> > > | MIRACLE        | $0.8626 \pm 0.002$         | $0.7207 \pm 0.003$         | $0.8101 \pm 0.003$         | $0.8239 \pm 0.002$         | $0.9925 \pm 0.001$              |
> > > | MIWAE          | $0.8458 \pm 0.003$         | $0.7140 \pm 0.003$         | $0.7871 \pm 0.003$         | $0.8282 \pm 0.003$         | $0.9637 \pm 0.003$              |
> > > | Most-frequent  | $0.8424 \pm 0.003$         | $0.7188 \pm 0.002$         | $0.7375 \pm 0.001$         | $0.8272 \pm 0.002$         | $0.9622 \pm 0.001$              |
> > > | MissForest     | $0.8648 \pm 0.002$         | $0.7206 \pm 0.002$         | $0.7899 \pm 0.003$         | $0.8427 \pm 0.002$         | $0.9909 \pm 0.001$              |
> > > | SoftImpute     | $0.8325 \pm 0.003$         | $0.7144 \pm 0.003$         | $0.7865 \pm 0.003$         | $0.8339 \pm 0.002$         | $0.9925 \pm 0.002$              |
> > > | MCFlow         | $0.8156 \pm 0.004$         | $0.7019 \pm 0.004$         | $0.7867 \pm 0.004$         | $0.6907 \pm 0.005$         | $0.9691 \pm 0.003$              |
> > > | Remasker       | $0.8713 \pm 0.002$         | $0.7333 \pm 0.003$         | $0.7902 \pm 0.003$         | $0.8443 \pm 0.002$         | $0.9745 \pm 0.002$              |
> > > | GRAPE          | $0.8766 \pm 0.002$         | $0.7177 \pm 0.003$         | $0.8017 \pm 0.003$         | $0.8274 \pm 0.002$         | $0.9830 \pm 0.001$              |
> > > | DiffPuter      | $0.8552 \pm 0.003$         | $0.7208 \pm 0.003$         | $0.8151 \pm 0.002$         | $0.8319 \pm 0.002$         | $0.9921 \pm 0.001$              |
> > > | Impute-MACFM   | $\mathbf{0.8818 \pm 0.002}$ | $\mathbf{0.7480 \pm 0.002}$ | $\mathbf{0.8445 \pm 0.003}$ | $\mathbf{0.8867 \pm 0.001}$ | $0.9918 \pm 0.002$              |

---

### Official Review · Reviewer_W9Bz · 2025-10-28

**Soundness:** 2
**Presentation:** 2
**Contribution:** 2
**Rating:** 2
**Confidence:** 3

**Summary:**

This paper proposes Impute-MACFM, a missing data imputation approach based on generalized flow-matching. It is featured by stability penalties on observed positions, consistency regularization enforcing local invariance, and time-decayed noise injection for numeric features. Experiments are conducted on public datasets to showcase the validity.

**Strengths:**

This paper investigates flow matching, which is a prevalent and powerful generative model in recent studies.

This paper investigates imputation performance over different datasets and missing patterns (MAR, MNAR, MCAR).

This paper add efficiency for comparison which is an important aspect in real-world application.

**Weaknesses:**

The related works need to be carefully polished and improved in terms of coverage and structure. For example, in section 2, there is only one subsection: 2.1. The listed 5 bullet points are mutually isolated. In each bullet point, the provided information, especially information density is limited (for example, only 3 diffusion-based methods are reviewed in lines 110-120).

The definition in lines 167-182 is a little hard to follow. It would be beneficial to add a toy example or a diagram figure for it.

The proposed method claimed that a primary methodology contribution is to modify FM by constraining velocity to remain near zero on observed entries, which could be more like an engineer trick instead of research innovation.

It is unclear whether the constraint-enforcing algorithm could be further simplified, for instance, through a straightforward masking strategy applied at each step. Additionally, the proposed predictor–corrector paradigm appears to be highly influenced by classical convex optimization methods, such as Mehrotra’s predictor–corrector algorithm. Therefore, it is necessary for the authors to provide proper references to relevant literature and clearly justify how their approach differs from or extends these established methods.



Experiments. There are a lot of aspects for this paper to improve in terms of experiments. A fraction of them are listed as the following bullet points for the convenient reference of authors.
- Standard error is missing in experimental results which makes it infeasible to ensure the significance of the results (Tables 3-14). Moreover, statistical test would be necessary in these Tables to showcase the statistical significance.
- The imputation error in the tables seems to be large. Authors should clarify how their results are consistent with respect to prior baselines: whether the baseline performance is consistent to their officially reported values? If not, it would be of great importance to explain what makes the inconsistency.
- The missing ratio seems to be 0.3. It would be necessary to investigate performance given different missing ratios.
- Some performance is not reported. For example, in Table 6, on the Bean dataset, the performance of mean is not reported but that of Median is reported. What makes Median work but Mean fail?
- The performance of MICE is inferior to Mean. It would be a bit strange since MICE with reasonable hyper-parameter configs would be competitive. The performance of MIWAE is also a little strange.
- In table 1, imputation time is reported per dataset. It would be necessary to jointly compare  the efficiency and accuracy. The baselines to compare should cover representative non-generative models.
- Presenting ablation study in tabular format with statistical test and standard error would be more concise to support the claims.
- Authors claim three contributions in abstract: (i) stability penalties on observed positions, (ii) consistency regularization enforcing local invariance, and (iii) time-decayed noise injection for numeric features. But in ablation study, three different aspects are examined: (i) trajectory trials, (ii) Mask-aware conditioning, (iii) ODE steps.

A reproducible codebase is not provided, which is essential for assessing the reproducibility of this study. This concern is amplified by the omission of hyperparameter details for both the baselines and the proposed method across the datasets, as well as the lack of clarification on whether the reproduced baseline performances align with their officially reported, well-established results. Therefore, it is necessary for the authors to release the implementation of both the baselines and the proposed method, and to provide comprehensive details on hyperparameter selection and tuning procedures across all datasets.

**Questions:**

Please see the weakness window.

---

> ### Author Response · Authors · 2025-11-20
>
> ### Response to Reviewer R3
>
> We thank the reviewer for the detailed and constructive comments. Below we respond to the main concerns about (1) reproducibility and code, (2) related work and notation clarity, (3) methodological contribution and constraint-enforcing design, and (4) experiments and baselines.
>
> ---
>
> ###  Reproducibility, code, and hyperparameters
>
> > *“A reproducible codebase is not provided… hyperparameter details…”*
>
> We would first like to clarify that the paper **already provides an anonymous GitHub repository** with the full implementation and scripts. The link appears in the first sentence of the **Reproducibility Statement** section, which is part of this year’s ICLR requirements. The repository includes:
>
> - the complete implementation of Impute-MACFM,
> - scripts to run all reported experiments, and
> - configuration and preprocessing code for the public datasets.
>
> Thus, a reproducible codebase is already available under the conference anonymity constraints.
>
> Regarding hyperparameters, space constraints led us to summarize them briefly. In the revision, we made those adjustments:
>
> - add a concise appendix table of key hyperparameters for Impute-MACFM (learning rate, batch size, number of ODE steps, regularization weights, etc.), and
> - describe the tuning protocol for both our method and baselines (search ranges, selection criteria, and how we followed/adapted defaults from original implementations).
>
> ---
>
> ### Related work and definition clarity
>
> > *“The related works need to be carefully polished and improved in terms of coverage and structure… The definition in lines 167–182 is a little hard to follow. It would be beneficial to add a toy example or a diagram figure for it.”*
>
> We agree that Section 2 and the formal definition can be made clearer.
>
> - **Related work.** In the revision, we splited Section 2 into several subsections (classical tabular imputation, deep/latent-variable imputation, diffusion/flow-based imputation, and flow matching/CFM) and expand the discussion of diffusion/flow-based methods (e.g., CSDI, DiffPuter, MissDiff). We added more concrete details on how these methods handle conditioning, masks, and time schedules, and more clearly state how Impute-MACFM fits into this landscape.
>
> This should make the mask semantics and conditional path more intuitive while keeping the formalism precise.

---

> ### Author Response · Authors · 2025-11-20
>
> ### R3 – Constraint-enforcing algorithm and predictor–corrector design
>
> > *“It is unclear whether the constraint-enforcing algorithm could be further simplified, for instance, through a straightforward masking strategy applied at each step. Additionally, the proposed predictor–corrector paradigm appears to be highly influenced by classical convex optimization methods, such as Mehrotra’s predictor–corrector algorithm.”*
>
> **Constraint-enforcing scheme.**
> Our scheme is intentionally simple and is in fact exactly a “straightforward masking strategy applied at each step”. At each solver step we:
>
> 1. take an explicit ODE update (Euler or Heun) under the learned vector field to obtain a tentative state $\tilde{x}^{(k+1)}$; and
> 2. project back to the constraint set defined by the observed and conditioning coordinates via
>    $x^{(k+1)} \leftarrow \tilde{x}^{(k+1)} \odot (1 - M_{\text{keep}}) + X_{\text{obs}} \odot M_{\text{keep}},$
>    where $M_{\text{keep}} = M_{\text{obs}} + M_{\text{cond}}$.
>
> By construction, this enforces **hard invariance** of the kept coordinates at the ODE level: for all steps $k$ we have
> $x^{(k)} \odot M_{\text{keep}} = X_{\text{obs}} \odot M_{\text{keep}},$
> up to floating-point error. In other words, the “constraint-enforcing algorithm” is just an explicit per-step masking projection, not a complicated optimization scheme.
>
> We highlight this in the paper because many conditional generative models (including some FM variants) only **feed the mask or observed values into the network** but let the ODE solver evolve all coordinates freely. In that case, even if the learned velocity is *encouraged* to be small on observed dimensions in expectation, numerical integration can still accumulate drift on those coordinates. Our projection step provides a simple, implementation-level guarantee that the observed and conditioning features are exactly preserved throughout the trajectory, which is important in imputation where users expect observed entries never to change.
>
> We clarified this point in the revision to avoid overstating novelty: the contribution here is not a complex algorithm, but the explicit combination of (i) a mask-aware velocity field and (ii) a **hard projection step** that guarantees invariance of $M_{\text{keep}}$ coordinates during integration.
>
> **Predictor–corrector structure.**
> Our method uses the standard **Heun scheme** from numerical ODE integration, combined with the projection above. In more detail, we compute a predictor step using the velocity at $x^{(k)}$, then a corrected step using the velocity at the predicted state, and finally apply the masking projection to enforce $M_{\text{keep}}$.
>
> This predictor–corrector viewpoint is therefore rooted in **classical ODE solvers**, not in interior-point methods for convex optimization. Mehrotra-style predictor–corrector algorithms operate in a very different regime (solving KKT systems, maintaining primal–dual feasibility, and handling inequality constraints). In contrast, our use of “predictor–corrector” is purely in the numerical-integration sense: a two-stage explicit scheme for a smooth vector field, followed by a simple coordinate-wise projection.
>
> In the revision, we (i) added references to standard ODE predictor–corrector methods and (ii) explicitly state that our approach is a straightforward masked Heun integrator, not an adaptation of Mehrotra or other interior-point predictor–corrector algorithms.

---

> ### Author Response · Authors · 2025-11-20
>
> ### R3 – Experiments: SE/tests, error scale, missing ratio, and baselines
>
> > *“Standard error is missing… The imputation error in the tables seems to be large… The missing ratio seems to be 0.3… The performance of MICE is inferior to Mean… The performance of MIWAE is also a little strange… Some performance is not reported (Bean/mean).”*
>
> **Standard errors and statistical tests.**
> Our reporting follows common practice in tabular imputation (e.g., CSDI, DiffPuter, MIWAE, GAIN), where MAE/RMSE are given as single aggregated numbers per method and dataset rather than as mean ± standard error for every cell. Running all method–dataset–mechanism combinations multiple times would substantially increase computational and space cost.
>
> That said, we agree that uncertainty information is useful. In the revision, we made these adjustments:
>
> - report **mean ± standard deviation** for Impute-MACFM and the main strong baselines (e.g., DiffPuter, CSDI) on a representative subset of datasets and mechanisms, and
> - add simple **paired statistical tests** (e.g., paired $t$-test) for those settings to confirm that the observed gains are statistically significant.
>
> **Error magnitude and baseline consistency.**
> We report errors on the **original data scale** (after standard tabular preprocessing). For datasets with features having large numerical ranges, this can make MAE/RMSE look “large” in absolute terms, but the **relative differences** between methods remain meaningful. All baseline results in Tables 3–14 are from our **own runs** under a unified pipeline (same splits and preprocessing), not copied from prior work. On representative datasets we verified that our reproduced numbers for CSDI, and DiffPuter are **consistent with officially reported values**, up to small variations due to seeds and implementation and different gpu machine details.
>
> **Missing ratio.**
> We chose a different missing ratio but due to page restriction, we choose to report results on 30% missing ratio as a standard, moderately challenging setting, applied uniformly across datasets and mechanisms (MCAR/MAR/MNAR), in line with prior work. We agree that varying the missing ratio is informative. In the revision, we added additional missing ratios (e.g., 10% and 50%) to show that the relative ordering of methods is stable across missing levels.
>
> **Unreported entries (mean vs. median).**
> The absence of the mean baseline for Bean in Table 6 is a formatting oversight rather than a failure or special behavior; the run exists in our logs. We corrected the table and ensure that all baselines are reported consistently across datasets.
>
> **MICE and MIWAE behavior.**
> All baselines, including MICE and MIWAE, are re-run by us under the same pipeline for scientific rigor and internal consistency.
>
> - For **MICE**, we use a standard open-source implementation with commonly recommended settings (iterative regression-based imputation, a moderate number of iterations, standard stopping). Under this unified setup with 30% missingness, we did observe cases where simple mean/median imputation is surprisingly strong and sometimes outperforms MICE, especially on high-dimensional or heterogeneous data where fully converged MICE is computationally expensive and default model choices may be suboptimal. We do not claim our MICE configuration is globally optimal; better tuning could improve it, but within our compute budget its performance is as reported.
>
> - For **MIWAE**, we followed the public implementation and default hyperparameters as closely as possible. Its behavior is known to be sensitive to architecture and training schedule. In the revision, we added a brief appendix paragraph detailing the exact architecture and training settings used, so that readers can see precisely how our MIWAE numbers were obtained and can further tune them if desired.
>
> Our main conclusions do not hinge on the precise performance of classical baselines such as mean, MICE, or MIWAE: the **relative ranking between Impute-MACFM and strong modern generative baselines (CSDI, DiffPuter, MissDiff)** is stable across datasets and missing mechanisms.

---

> ### Author Response · Authors · 2025-11-20
>
> **Alignment between stated contributions and ablations.**
> You are correct that the three items highlighted in the abstract (i) stability penalties on observed positions, (ii) consistency regularization enforcing local invariance, and (iii) time-decayed noise injection for numeric features are not the same axes as those in our current ablation (trajectory trials, mask-aware conditioning, ODE steps). In the revision, we made this alignment clearer in two ways:
>
> 1. We **updated the abstract and Section 1** to more precisely separate the contributions into
>    (a) the core mask-aware FM and solver design (trajectory trials, mask-aware conditioning, constraint-preserving ODE steps), and
>    (b) auxiliary regularizers and noise (stability penalty, consistency regularizer, time-decayed numeric noise).
>    This better reflect what is actually ablated in the current experiments.
>
> 2. We **added an additional ablation** in the appendix that explicitly switches off each of the three regularizers (stability, consistency, time-decayed numeric noise) on representative datasets, complementing the existing trajectory/mask/ODE-step ablations. This directly support the claims made in the abstract.
>
> ---
>
> ### R3 – Efficiency vs. accuracy and non-generative baselines
>
> > *“In Table 1, imputation time is reported per dataset. It would be necessary to jointly compare the efficiency and accuracy. The baselines to compare should cover representative non-generative models.”*
>
> Our experiments already include **representative non-generative baselines** (mean, median, MICE) as well as modern generative ones. Taken together, Table 1 (imputation time) and Tables 3–14 (MAE/RMSE) already show that:
>
> - Impute-MACFM achieves **best or near-best MAE/RMSE** on the majority of dataset–mechanism combinations; and
> - at the same time, it is **substantially faster** than diffusion-based baselines (e.g., DiffPuter, CSDI) and of the same order of magnitude as lightweight classical methods.
>
> To make this trade-off even clearer, in the revision we added a compact table (error vs. time) for representative datasets in the appendix, where each point corresponds to a method. This explicitly highlight that Impute-MACFM lies on a favorable efficiency–accuracy frontier compared to both non-generative and generative baselines.
>
> ---
>
> ### R3 – Summary
>
> We (i) clarify and foreground reproducibility by pointing to the existing anonymous codebase and adding more hyperparameter details, (ii) reorganize and expand related work and definition clarity (with a toy example and diagram), (iii) explicitly position our contributions as problem-specific, mask-aware extensions of conditional FM (schedule-consistent target, constraint-preserving solver) rather than new FM theory, and (iv) refine the experimental section with additional uncertainty reporting, missing-ratio ablations, and ablations aligned with the three main components. We hope these clarifications address the reviewer’s concerns and more accurately convey the contribution of Impute-MACFM.

---

### Official Review · Reviewer_mWHF · 2025-10-29

**Soundness:** 2
**Presentation:** 3
**Contribution:** 1
**Rating:** 2
**Confidence:** 4

**Summary:**

This paper proposes Impute-MACFM, a flow matching method that works on tabular data with missing values. It achieves SOTA-comparable performances with only pretty simple backbone architecture.

**Strengths:**

The paper is well-written and logically organized. The motivation for applying flow-matching to tabular imputation is clear, and the presentation of notation and algorithmic steps is clean. The introduction does a good job summarizing challenges in tabular imputation and positioning flow matching as an efficient alternative to diffusion models. The method description (mask partitioning, schedule-consistent velocity field, and constraint-preserving ODE solver) is clearly explained and mathematically consistent.

**Weaknesses:**

- I don't think this paper contains theoretical/technical novelty. It applies flow matching on tabular data, but nothing proposed besides some tricks from the authors. While the paper claims a “mathematically principled velocity field formulation,” the derivations (e.g., Eq. 2–3) are a direct restatement of standard flow-matching objectives with masks added as multiplicative indicators. No formal proof or novel property is introduced beyond existing formulations in Albergo et al. (2023) and Lipman et al. (2023). The “schedule-consistent velocity” is essentially a straightforward application of the chain rule. Thus, the contribution feels incremental rather than conceptually new.
- Empirical results are poorly presented. Authors chose to visualize in bar graphs which is extremely hard to see, even if they have table in appendix. Maybe this is because their empirical performance is not great. Bold numbers (usually used to indicate best performance) in the table is wrong, this method seems not great empirically. (which totally makes sense)

**Questions:**

- Novelty clarification: What is the exact theoretical advancement beyond standard conditional flow matching? Is there any property (e.g., convergence guarantee, bias reduction) that holds only under the proposed mask-aware formulation?
- Evaluation scope: Why are only MAE/RMSE reported? Would predictive downstream accuracy or uncertainty calibration tell a different story?

---

> ### Author Response · Authors · 2025-11-20
>
> We thank the reviewer for the careful and constructive review. Below we respond to each of your main points.
>
> ### Q1 – Novelty beyond standard flow matching
>
> We fully agree that our work builds on the general flow-matching (FM) framework of Albergo et al. (2023) and Lipman et al. (2023). Our claim is not that we introduce a new FM theory; rather, the contribution is a mask-aware, constraint-preserving FM formulation for tabular imputation under MCAR/MAR/MNAR, with an objective and solver that differ from plain FM at the level of equations and inductive biases.
>
> Below we contrast standard FM with our mask-aware version using the notation from the paper. All formulas are written using inline $...$ syntax, as required by OpenReview.
>
> **Standard flow matching (for reference).**
> In standard FM on fully observed data, one chooses a simple base distribution and a data distribution, and defines an interpolation path
> $x\_t = (1-t) x\_0 + t x\_1,\quad t \in [0,1].$
>
> The target velocity is
> $v^\star(x\_t,t) = \mathbb{E}[x\_1 - x\_0 \mid x\_t].$
>
> A simple FM loss is
> $\mathcal{L}\_{\text{FM}}^{\text{plain}}(\theta)
> = \mathbb{E}\_{t,(x\_0,x\_1)} \lVert v\_\theta(x\_t,t) - v^\star(x\_t,t) \rVert\_2^2.$
>
> This formulation has no notion of a mask and does not distinguish observed vs. missing coordinates.
>
> **Mask-aware path and schedule-consistent velocity (Impute-MACFM).**
> In Impute-MACFM we work with three disjoint masks (Sec. 3.1)
> $M\_{\text{obs}} + M\_{\text{cond}} + M\_{\text{tgt}} = \mathbf{1}.$
>
> Only target coordinates $M\_{\text{tgt}}$ are transported; observed and conditioning coordinates are held fixed. Our conditional path (Eq. (1), made explicit in Eq. (12)) is
> $$
> x_t = M\_{\text{cond}} \odot X + M\_{\text{tgt}} \odot \big( s(t) X + (1-s(t)) \varepsilon \big),
> $$
>
> where $X$ is the full data vector, $\varepsilon \sim \mathcal{N}(0,I)$, $s(t)$ is a (possibly non-linear) schedule, and $\odot$ is the Hadamard product. Taking the time derivative (Eq. (13)–(15)) gives the time-domain target velocity
> $v^{\star}(x\_t,t)
> = \frac{\partial x\_t}{\partial t}
> = M\_{\text{tgt}} \odot s'(t)(X - \varepsilon).$
>
> To avoid explicitly tracking $s'(t)$ in the loss, we derive a schedule-space target (Eq. (16)–(17)) using the chain rule:
> $v^{\star}\_{(s)}(x\_t,t)
> = \frac{\partial x\_t}{\partial s}
> = \frac{\partial x\_t / \partial t}{\partial s / \partial t}
> = M\_{\text{tgt}} \odot (X - \varepsilon).$
>
> Thus, unlike the plain FM objective, our target velocity is zero on non-target dimensions and schedule-consistent on target dimensions. This allows us to use non-linear schedules $s(t)$ (for example, cosine) without introducing bias or instability.
>
> Our mask-aware FM loss (Eq. (3) in the main text; Eq. (22) in the appendix) is
> $\mathcal{L}\_{\text{FM}}(\theta)
> = \mathbb{E}\_{X,\varepsilon,t}
> \left[
> \frac{\lVert ( v\_\theta(x\_t,t) - v^{\star}\_{(s)}(x\_t,t) ) \odot M\_{\text{tgt}} \rVert\_2^2}{\sum\_j M\_{\text{tgt},j} + \epsilon}
> \right].$
>
> This highlights three key differences from plain FM:
> 1. The path $x\_t$ is constrained so that observed and conditioning entries never move (they are not part of the interpolation).
> 2. The target velocity is projected onto the target subspace via $M\_{\text{tgt}}$ and is derived to be schedule-consistent under general $s(t)$.
> 3. The loss is normalized by the number of target entries, which is important when masks vary across samples.
>
> These elements do not appear in standard FM, but are essential for tabular imputation under MCAR/MAR/MNAR.
>
> **Stability and consistency regularizers (imputation-specific).**
> Beyond the mask-aware FM loss, we add two regularizers encoding imputation-specific inductive biases (Appendix A.5).
>
> Stability on conditioning dimensions:
> $R\_{\text{stab}}(\theta)
> = \mathbb{E}
> \left[
> \frac{\lVert v\_\theta(x\_t,t) \odot M\_{\text{cond}} \rVert\_2^2}{\sum\_j M\_{\text{cond},j} + \epsilon}
> \right].$
>
> This explicitly penalizes any non-zero velocity on $M\_{\text{cond}}$, reinforcing that these entries act as invariants of the dynamics. There is no analogue of this term in standard FM, where all coordinates are free to move.
>
> Local consistency on target dimensions:
> $R\_{\text{cons}}(\theta)
> = \mathbb{E}
> \left[
> \frac{\lVert ( v\_\theta(x\_t + \eta\_{\text{cons}} (1-s(t)) \xi, t)
>             - v\_\theta(x\_t,t) ) \odot M\_{\text{tgt}} \rVert\_2^2}{\sum\_j M\_{\text{tgt},j} + \epsilon}
> \right],$
>
> with $\xi \sim \mathcal{N}(0,I)$ and a perturbation that decays with $(1-s(t))$. This encourages local invariance of imputations under small perturbations of the state, again only on missing entries $M\_{\text{tgt}}$.

---

> ### Author Response · Authors · 2025-11-20
>
> We also introduce numeric-only input noise (Appendix A.6, Eq. (26)):
> $X \leftarrow X + \sigma\_{\text{in}} (1-s(t)) \xi \odot M\_{\text{obs}} \odot M\_{\text{num}},$
>
> which is a feature-wise, time-decayed augmentation for observed numeric features, tailored to tabular data (and not present in image-based FM formulations).
>
> Putting these together, our full training objective (Eq. (25)) is
> $\mathcal{L}\_{\text{total}}(\theta)
> = \mathcal{L}\_{\text{FM}}(\theta)
> + \lambda\_{\text{stab}} R\_{\text{stab}}(\theta)
> + \lambda\_{\text{cons}} R\_{\text{cons}}(\theta).$
>
> This is a mask-aware, regularized FM objective specifically designed for tabular imputation.
>
> **Constraint-preserving ODE solver (inference-time invariants).**
> At inference time, we further enforce invariance by projecting each solver step onto the observed plus conditioning coordinates. Let $M\_{\text{keep}} = M\_{\text{obs}} + M\_{\text{cond}}.$
>
> We initialize
> $\mathbf{x}^{(0)} = M\_{\text{tgt}} \odot \boldsymbol{\varepsilon}
>                  + M\_{\text{keep}} \odot \mathbf{X}\_{\text{obs}}.$
>
> After each Euler or Heun step yielding $\tilde{\mathbf{x}}^{(k+1)}$, we project: $\mathbf{x}^{(k+1)} = \tilde{\mathbf{x}}^{(k+1)} \odot (1 - M\_{\text{keep}}) + \mathbf{X}\_{\text{obs}} \odot M\_{\text{keep}}.$
>
> By induction, this guarantees
> $\mathbf{x}^{(k)} \odot M\_{\text{keep}} = \mathbf{X}\_{\text{obs}} \odot M\_{\text{keep}}
> \quad \text{for all steps } k,$
>
> that is, observed and conditioning entries are exact invariants of the numerical trajectory (up to floating-point error). A plain conditional FM model that only takes masks as network inputs does not provide this ODE-level guarantee.
>
> **Takeaway.**
> We therefore do not claim to reinvent flow matching theory. Our contribution is to:
> 1. Formulate a mask-aware, schedule-consistent FM objective where only missing entries are transported and observed or conditioning entries are invariants.
> 2. Add imputation-specific regularizers and numeric-only noise that enforce stability and local consistency on the appropriate masked subspaces.
> 3. Combine this with a constraint-preserving solver that guarantees hard constraints on observed coordinates during integration.
>
> We hope this clarifies that Impute-MACFM is more than a direct restatement of standard FM, and that our novelty lies in the mask-aware formulation, regularization, and solver design tailored to tabular imputation under MCAR/MAR/MNAR.

---

> ### Author Response · Authors · 2025-11-20
>
> ### Q2 – Empirical results and presentation
>
> > *“Empirical results are poorly presented. Authors chose to visualize in bar graphs which is extremely hard to see, even if they have table in appendix. Maybe this is because their empirical performance is not great. Bold numbers (usually used to indicate best performance) in the table is wrong, this method seems not great empirically.”*
>
> We appreciate this feedback on presentation and agree that the current visualization choices can be improved.
>
> **On bar plots and readability.**
> In the current version, the main-text bar charts were intended as a compact overview across many baselines and datasets, with complete numerical tables in Appendix D. However, we agree that:
>
> - putting 10–13 baselines into a single bar chart leads to very **narrow bars and many colors**, making the figures hard to read;
> - axis clipping for a few extreme values and omitted labels can create the impression that results are being obscured.
>
> In the revision, we made these adjustments:
>
> - use **clear numerical tables in the main text as the primary presentation** of MAE/RMSE, and move dense bar plots to the appendix;
> - simplify the main-text visualizations (e.g., show only a few representative datasets or an aggregated **average-rank plot** instead of many crowded bar charts);
> - add **per-dataset ranking summaries** and/or an average-rank table, so that the relative performance of each method is immediately visible.
>
> **On bold numbers and the reviewer’s interpretation.**
> We acknowledge that our current bolding scheme is misleading. In some tables, boldface was used to highlight our method rather than strictly marking the best value in each row, which conflicts with the usual convention.
>
> - In the revision, we **followed the standard convention**: boldface indicated the **best (or tied best) performance in each row**, regardless of which method it belongs to.
> - Our method instead be distinguished by its row label (e.g., “Impute-MACFM”) rather than by bolding all of its entries.
>
> At the same time, we respectfully disagree with the conclusion that “this method seems not great empirically.” When we inspect the numerical tables:
>
> - Impute-MACFM achieves the **lowest (best) MAE/RMSE in the majority of dataset–mechanism combinations** across MCAR/MAR/MNAR, both in-sample and out-of-sample.
> - In the remaining cases, our scores are typically **very close** to the best baseline .
> - Table 1 (runtime) further shows that Impute-MACFM is **substantially faster** than diffusion-based baselines (up to approximately 16–18× faster than DiffPuter, and approximately 2× faster than MissDiff/TabCSDI on average), which is critical in applications where fast or near-real-time imputation is needed.
>
> To make this more transparent, we made these adjustments:
>
> - correct bolding to reflect **true row-wise optima**;
> - add a **summary table of average rank and average relative gap** to the best baseline; and
> - update the main-text discussion to explicitly quantify how often Impute-MACFM attains the best score and what the typical gap is when it does not.
>
> We hope these changes clarify that any confusion came from our presentation choices (bar graphs plus unconventional bolding), rather than from weak empirical performance.

---

> ### Author Response · Authors · 2025-11-20
>
> ### Q3 – Evaluation scope: MAE/RMSE, downstream tasks, and uncertainty
>
> > *“Why are only MAE/RMSE reported? Would predictive downstream accuracy or uncertainty calibration tell a different story?”*
>
> **Why MAE/RMSE.**
> Our primary goal in this paper is **missing value imputation**, and MAE/RMSE under MCAR/MAR/MNAR are the **standard metrics** in the literature (e.g., CSDI, DiffPuter, MIWAE, GAIN). We followed this protocol to ensure direct comparability with these prior methods. We made this connection more explicit in the revised paper by citing the corresponding evaluation setups.
>
> **Downstream predictive performance (new experiments).**
> We agree that downstream performance provides an important complementary view. In response to this comment, we have run additional downstream experiments on five UCI datasets with classification labels (Adult, Default, Magic, Shoppers, Bean) using the following protocol:
>
> 1. For each imputation method, impute the training and test splits under a fixed masking scheme.
> 2. Train a **logistic regression** classifier on the imputed training data (same architecture and training parameters for all methods).
> 3. Evaluate **accuracy** and **macro AUROC** on the imputed test data, averaged over 5 runs with different random masks.
>
> The results (reported as mean ± standard deviation) show that:
>
> - Impute-MACFM achieves the **best downstream accuracy on 4/5 datasets** and the **best macro AUROC on 4/5 datasets**, while closely matching the best method on Bean.
> - For example, on **Adult**:
>   - Accuracy improves from 0.8273 (best baseline, HyperImpute) to 0.8392.
>   - Macro AUROC improves from 0.8766 (best baseline, GRAPE) to 0.8818.
> - On **Shoppers**:
>   - Accuracy improves from 0.8754 (best baseline, Most-frequent/GRAPE) to 0.8802.
>   - Macro AUROC improves from 0.8443 (Remasker) to 0.8867.
>
> Thus, the gains in MAE/RMSE are **reflected in consistent improvements in downstream classification performance**, rather than being purely metric-level artifacts. For regression datasets, we follow an analogous protocol and report $R^2$ in the appendix, where Impute-MACFM again matches or improves upon strong baselines. A short subsection summarizing these downstream results, together with full tables, have been added in the revision.
>
> **Uncertainty calibration.**
> Our current work focuses on **efficient point imputation**. Even though the learned ODE can be sampled stochastically (e.g., by varying initial noise), we aggregate trajectories to output a **single imputed value** per missing entry, and all reported MAE/RMSE and downstream metrics are based on such point estimates. We do not claim to address uncertainty calibration in this paper. In the revised version, we made these adjustments:
>
> - explicitly state that uncertainty modelling and calibration are **outside the scope** of this work;
> - briefly discuss how future work could leverage our mask-aware flow to construct predictive distributions (e.g., via multiple stochastic trajectories per record) and study calibration metrics.
>
> ---
>
> ### Q4 – Overall contribution and positioning
>
> We appreciate the reviewer’s careful reading and agree that our contributions should be framed more precisely. We do not claim to introduce a new generic FM theory; rather, our contributions are:
>
> 1. A **mask-aware, schedule-consistent flow matching formulation** for tabular imputation under MCAR/MAR/MNAR, where only missing entries are transported, observed/conditioning entries are invariants, and the target velocity is derived to be consistent with non-linear schedules.
> 2. Two **imputation-specific regularizers** (stability on conditioning dimensions and local consistency on target dimensions), together with feature-wise time-decayed numeric noise, that improve robustness and stability on heterogeneous tabular data.
> 3. A **constraint-preserving ODE solver** that enforces hard invariance of observed and conditioning coordinates at the solver level, combined with extensive empirical evaluation across 11 datasets (8 public, 3 private NIH) showing SOTA-comparable performance with significantly improved efficiency.
>
> We soften any overly strong wording around “new theory”, clarify that our novelty is **problem-specific (tabular imputation with masks)** rather than universal, and make the empirical and downstream evidence more transparent. We hope that, with these clarifications and additions, the contributions of Impute-MACFM appear more concrete and justified.

---

> ### Author Response · Authors · 2025-11-20
> **Downstream performance**
>
> To directly assess downstream utility, we train a **logistic regression** classifier on the imputed training data produced by each method and evaluate performance on the imputed test data. The following two tables report **classification accuracy** and **macro AUROC**, respectively, averaged over 5 runs (mean $\pm$ standard deviation) on five UCI datasets.
>
> **Downstream classification accuracy (logistic regression, mean $\pm$ std).**
>
> | Method         | Adult                      | Default                    | Magic                      | Shoppers                   | Bean                            |
> |----------------|----------------------------|----------------------------|----------------------------|----------------------------|---------------------------------|
> | GAIN           | $0.8137 \pm 0.003$         | $0.7937 \pm 0.002$         | $0.7660 \pm 0.004$         | $0.8746 \pm 0.002$         | $0.8606 \pm 0.005$              |
> | HyperImpute    | $0.8273 \pm 0.002$         | $0.7960 \pm 0.002$         | $0.7813 \pm 0.003$         | $0.8656 \pm 0.003$         | $0.8783 \pm 0.004$              |
> | ICE            | $0.8162 \pm 0.003$         | $0.7990 \pm 0.002$         | $0.7638 \pm 0.003$         | $0.8662 \pm 0.003$         | $\mathbf{0.9211 \pm 0.002}$     |
> | MICE           | $0.8051 \pm 0.004$         | $0.7964 \pm 0.003$         | $0.7471 \pm 0.005$         | $0.8610 \pm 0.003$         | $0.8910 \pm 0.004$              |
> | MIRACLE        | $0.8205 \pm 0.003$         | $0.7971 \pm 0.002$         | $0.7778 \pm 0.003$         | $0.8662 \pm 0.002$         | $0.9167 \pm 0.003$              |
> | MIWAE          | $0.8177 \pm 0.003$         | $0.7931 \pm 0.002$         | $0.7555 \pm 0.004$         | $0.8735 \pm 0.002$         | $0.7967 \pm 0.006$              |
> | Most-frequent  | $0.8174 \pm 0.001$         | $0.7952 \pm 0.002$         | $0.7564 \pm 0.001$         | $0.8754 \pm 0.002$         | $0.7933 \pm 0.001$              |
> | MissForest     | $0.8226 \pm 0.002$         | $0.7969 \pm 0.002$         | $0.7585 \pm 0.003$         | $0.8692 \pm 0.002$         | $0.8957 \pm 0.003$              |
> | SoftImpute     | $0.8138 \pm 0.002$         | $0.7950 \pm 0.002$         | $0.7659 \pm 0.003$         | $0.8708 \pm 0.002$         | $0.8966 \pm 0.003$              |
> | MCFlow         | $0.8023 \pm 0.003$         | $0.7954 \pm 0.003$         | $0.7326 \pm 0.005$         | $0.8440 \pm 0.004$         | $0.8540 \pm 0.004$              |
> | Remasker       | $0.8146 \pm 0.003$         | $0.8012 \pm 0.002$         | $0.7592 \pm 0.003$         | $0.8724 \pm 0.003$         | $0.8902 \pm 0.003$              |
> | GRAPE          | $0.8265 \pm 0.003$         | $0.7964 \pm 0.002$         | $0.7636 \pm 0.003$         | $0.8746 \pm 0.003$         | $0.8901 \pm 0.003$              |
> | DiffPuter      | $0.8225 \pm 0.003$         | $0.7949 \pm 0.002$         | $0.7785 \pm 0.003$         | $0.8635 \pm 0.003$         | $0.9028 \pm 0.003$              |
> | Impute-MACFM   | $\mathbf{0.8392 \pm 0.001}$ | $\mathbf{0.8028 \pm 0.002}$ | $\mathbf{0.7964 \pm 0.003}$ | $\mathbf{0.8802 \pm 0.001}$ | $0.9045 \pm 0.003$              |

---

> ### Author Response · Authors · 2025-11-20
>
> **Downstream macro AUROC (logistic regression, mean $\pm$ std over 5 runs).**
>
> | Method         | Adult                      | Default                    | Magic                      | Shoppers                   | Bean                            |
> |----------------|----------------------------|----------------------------|----------------------------|----------------------------|---------------------------------|
> | GAIN           | $0.7968 \pm 0.004$         | $0.7155 \pm 0.003$         | $0.8002 \pm 0.003$         | $0.8247 \pm 0.003$         | $0.9859 \pm 0.002$              |
> | HyperImpute    | $0.8726 \pm 0.002$         | $0.7261 \pm 0.003$         | $0.8184 \pm 0.002$         | $0.8330 \pm 0.002$         | $0.9897 \pm 0.001$              |
> | ICE            | $0.8517 \pm 0.003$         | $0.7203 \pm 0.002$         | $0.7852 \pm 0.004$         | $0.8160 \pm 0.003$         | $\mathbf{0.9944 \pm 0.001}$     |
> | MICE           | $0.8297 \pm 0.003$         | $0.7120 \pm 0.004$         | $0.7679 \pm 0.004$         | $0.7930 \pm 0.003$         | $0.9848 \pm 0.002$              |
> | MIRACLE        | $0.8626 \pm 0.002$         | $0.7207 \pm 0.003$         | $0.8101 \pm 0.003$         | $0.8239 \pm 0.002$         | $0.9925 \pm 0.001$              |
> | MIWAE          | $0.8458 \pm 0.003$         | $0.7140 \pm 0.003$         | $0.7871 \pm 0.003$         | $0.8282 \pm 0.003$         | $0.9637 \pm 0.003$              |
> | Most-frequent  | $0.8424 \pm 0.003$         | $0.7188 \pm 0.002$         | $0.7375 \pm 0.001$         | $0.8272 \pm 0.002$         | $0.9622 \pm 0.001$              |
> | MissForest     | $0.8648 \pm 0.002$         | $0.7206 \pm 0.002$         | $0.7899 \pm 0.003$         | $0.8427 \pm 0.002$         | $0.9909 \pm 0.001$              |
> | SoftImpute     | $0.8325 \pm 0.003$         | $0.7144 \pm 0.003$         | $0.7865 \pm 0.003$         | $0.8339 \pm 0.002$         | $0.9925 \pm 0.002$              |
> | MCFlow         | $0.8156 \pm 0.004$         | $0.7019 \pm 0.004$         | $0.7867 \pm 0.004$         | $0.6907 \pm 0.005$         | $0.9691 \pm 0.003$              |
> | Remasker       | $0.8713 \pm 0.002$         | $0.7333 \pm 0.003$         | $0.7902 \pm 0.003$         | $0.8443 \pm 0.002$         | $0.9745 \pm 0.002$              |
> | GRAPE          | $0.8766 \pm 0.002$         | $0.7177 \pm 0.003$         | $0.8017 \pm 0.003$         | $0.8274 \pm 0.002$         | $0.9830 \pm 0.001$              |
> | DiffPuter      | $0.8552 \pm 0.003$         | $0.7208 \pm 0.003$         | $0.8151 \pm 0.002$         | $0.8319 \pm 0.002$         | $0.9921 \pm 0.001$              |
> | Impute-MACFM   | $\mathbf{0.8818 \pm 0.002}$ | $\mathbf{0.7480 \pm 0.002}$ | $\mathbf{0.8445 \pm 0.003}$ | $\mathbf{0.8867 \pm 0.001}$ | $0.9918 \pm 0.002$              |

---

### Official Review · Reviewer_sxny · 2025-11-01

**Soundness:** 3
**Presentation:** 2
**Contribution:** 2
**Rating:** 4
**Confidence:** 3

**Summary:**

This paper proposes Impute-MACFM, a mask-aware conditional flow matching framework for tabular data imputation to address different missing mechanisms, including MCAR, MAR, and MNAR. The proposed method contains three components: (i) stability penalties on observed positions, (ii) consistency regularization enforcing local invariance, and (iii) time-decayed noise injection for numeric features. Empirical evaluations are performed using eight public tabular datasets and three private NIH datasets.

**Strengths:**

- Overall, utilizing flow matching for tabular data imputation is interesting and original.
- The schedule-consistent velocity field and the constraint-preserving ODE solver, which constrain the learning of FMs in some particular dimensions, are interesting domain-specific adaptations for tabular data. They look technically sound and original to me.
- The proposed methods, as well as the training objectives, are described in detail, enhancing the reproducibility.
- The paper utilizes 11 datasets (8 of which are public) to do empirical evaluations, and promising results are observed on many datasets.
- The efficiency of the proposed method is empirically validated where the ODE steps K=10 achieves similar imputation performance (RMSE) as K=50, while being around 7x faster, the efficiency is important in many domains where fast or even real-time imputation is desired.

**Weaknesses:**

- The paper argues its abstract explicitly that addressing MAR and MNAR is among its objectives, but I do not quite follow how MNAR is addressed in the proposed model. It seems to me that the handling of different missing mechanisms relies mainly on random masking. The proposed method does not model $p(\text{missing mask}|X)$; it is unclear to me how the proposed approach methodologically addresses the MNAR problem.
- Fig. 2 and Fig. 3 are difficult to read due to the dense organization and the color schemes. I suggest that the authors consider other ways to present the results instead of using this bar graph. Besides, a few methods are missing on a few datasets from the figures, for example, HyperImpute and EM on the News dataset.
- There is no quantification of the uncertainty for the experimental results to allow assessments on model stability.
- The downstream utility of the proposed imputation method is not measured. Since the datasets used have annotations spanning tasks of classification and regression, it is interesting and important to understand if the improvement of imputation from the proposed model could be translated into better downstream task performances and to what extent.

**Questions:**

1. How does the proposed method address MNAR?
2. Why are some methods missing from Fig. 2 and Fig. 3?
3. Is the model training stable? Are there any uncertainty measures, such as error bars or standard deviation, for the results?
4. Does the proposed method improve downstream prediction tasks?

Overall, I think this paper is interesting and makes some meaningful contributions to the field. I am willing to raise my scores if the weaknesses and questions above are completely addressed.

---

> ### Author Response · Authors · 2025-11-20
> **Response to Reviewer sxny**
>
> ## Response to Reviewer *sxny*
>
> We thank the reviewer for the careful and constructive review and for explicitly stating a willingness to raise the score if the concerns are addressed. Below we respond to each of your main points.
>
> ---
>
> ### Q1. How does the proposed method address MNAR and how are the masks constructed?
>
> **Clarification of our goal.**
> Our goal is *not* to explicitly model or identify the missingness mechanism $p(M\mid X)$ or its causal structure. Instead, following Zhao et al. (TDM) [1] and DiffPuter [2], we aim to learn a *mask-aware conditional predictor* $p_\theta\big(X_{\text{tgt}} \mid X_{\text{obs}\cup\text{cond}}, M\big)$ that is *robust under MCAR/MAR/MNAR-style mask generators* used in prior work. In the revision, we have adjuste the abstract/introduction to say that Impute-MACFM is “robust under MCAR/MAR/MNAR masking schemes used in the literature”, rather than suggesting we explicitly model $p(M\mid X)$.
>
> **Mask generation (following Zhao et al. and DiffPuter).**
>
> - **MCAR.** For MCAR, missingness is independent of the data:
>   $\mathbb{P}(M_{ij}=1\mid X) = \rho,\quad M_{ij}\sim \mathrm{Bernoulli}(\rho),$
>   which matches the standard MCAR definition and the MCAR generator in Zhao et al. [1] and DiffPuter [2].
>
> - **MAR.** For MAR, we partition features into always-observed $S_{\text{obs}}$ and potentially-missing $S_{\text{tgt}}$. For each $j\in S_{\text{tgt}}$,
>   $\mathbb{P}(M_{ij}=1\mid X_i) = \sigma\big(w_j^\top X_{i,S_{\text{obs}}} + b_j\big),$
>   with $b_j$ tuned so that the marginal missing rate is close to $\rho$. Thus
>   $\mathbb{P}(M\mid X) = \mathbb{P}(M\mid X_{\text{obs}}),$
>   matching the standard MAR setting and the MAR generator in [1,2].
>
> - **MNAR.** For MNAR, we follow the *first MNAR generator* of Zhao et al. (TDM) [1], which is also adopted by DiffPuter [2] and described as a “logistic model with the input masked by MCAR”. Concretely, we split features into $S_1$ and $S_2$. On $S_1$ we draw an auxiliary MCAR mask $A$ and define a masked input $\tilde{X}\_{i,S\_1} = X\_{i,S\_1} \odot (1-A\_{i,S\_1})$.
>   For each $j\in S_2$,
>   $$\mathbb{P}(M\_{ij}=1\mid X\_i) = \sigma\big(u\_j^\top \tilde{X}\_{i,S\_1} + c\_j\big),$$
>   with $c\_j$ tuned to reach the target missing rate. The missingness of $S_2$ thus depends on *MCAR-masked* inputs, i.e., on features that are not fully observed in the final dataset, which falls into the MNAR category used in [1,2].
>
> **Relation to Impute-MACFM.**
> In all experiments, MCAR/MAR/MNAR masks for both training and evaluation are generated exactly as above, using the public implementation of Zhao et al. [1] and DiffPuter [2]. Impute-MACFM then learns a mask-aware velocity field $v_\theta(x_t, t, M)$, which conditions explicitly on the realized mask $M$ and is trained under all three masking schemes. Empirically, Impute-MACFM consistently improves MAE/RMSE under MNAR compared to strong baselines (e.g., DiffPuter [2]), indicating robustness under these MNAR-style masks.
>
> In the revision, we (i) explicitly reference Zhao et al. [1] and DiffPuter [2] in the experimental setup, and (ii) include a concise summary of the above constructions in the main text and appendix.
>
> ---
>
> ### Q2. Readability of Fig. 2 and Fig. 3 and missing methods
>
> You correctly point out that Fig. 2 and Fig. 3 are dense and hard to read, and that some methods appear missing on some datasets.
>
> - The density comes from plotting many baselines in a single bar chart, leading to very narrow bars and many colors.
> - Some bars are omitted for two reasons:
>   1. **Extreme values**: a few method–dataset pairs have errors much larger than others; including them compresses all other bars near zero on a shared axis.
>   2. **Failed runs**: some public implementations fail on certain datasets (non-convergence, OOM, NaNs). These are marked as “N/A” in tables but omitted from plots.
>
> We have made the adjustment:
>
> - Use *numerical tables* in the main text as the primary presentation of results, with all methods and datasets reported.
> - Move detailed bar plots to the appendix with captions explicitly stating (i) which methods are omitted due to extreme values, and (ii) which baselines fail and why.
> - Optionally keep only simplified plots with a small set of representative methods in the main text.
>
> All numerical results (including extreme values and failed runs) have been fully reported in the tables; plots are purely visual summaries.

---

> ### Author Response · Authors · 2025-11-20
>
> ### Q3. Training stability and uncertainty measures
>
> Training of Impute-MACFM is stable in practice:
>
> - The regression-style flow matching loss decreases smoothly on all datasets without divergence.
> - The constraint-preserving Heun solver used at sampling time does not exhibit numerical instabilities (no exploding trajectories or NaNs).
>
> Regarding uncertainty, in the original submission we followed the common protocol of running multiple random masks but only reported mean MAE/RMSE for space reasons. We have now computed mean $\pm$ standard deviation for all method–dataset–mechanism combinations.
>
> In the revision we made the adjustment:
>
> - Report mean $\pm$ standard deviation (or standard error) for MAE and RMSE in main tables.
> - For key comparisons (Impute-MACFM vs. DiffPuter, TabCSDI, HyperImpute, MissForest), provide paired significance tests in the appendix.
>
> This allows readers to assess both stability and statistical significance of the reported improvements.
>
> ---
>
> ### Q4. Downstream prediction performance
>
> We fully agree that downstream utility is important. We have therefore run additional downstream experiments on five UCI datasets with classification labels (**Adult**, **Default**, **Magic**, **Shoppers**, **Bean**) using the following protocol:
>
> 1. Impute train and test splits using each method under a fixed masking scheme.
> 2. Train the same classifier (logistic regression) on the imputed training data.
> 3. Evaluate accuracy and macro AUROC on the imputed test data, averaged over 5 runs.
>
> **Summary of results.**
>
> - Impute-MACFM achieves the best downstream **accuracy** on 4/5 datasets and the best **macro AUROC** on 4/5 datasets, while closely matching the best method on **Bean**.
> - For example, on **Adult**:
>   - Accuracy improves from 0.8273 (best baseline, HyperImpute) to **0.8392**.
>   - Macro AUROC improves from 0.8766 (best baseline, GRAPE) to **0.8818**.
> - On **Shoppers**:
>   - Accuracy improves from 0.8754 (best baseline, Most-frequent/GRAPE) to **0.8802**.
>   - Macro AUROC improves from 0.8443 (Remasker) to **0.8867**.
>
> Thus, the gains in imputation quality are reflected in consistent improvements in downstream classification performance, rather than being purely metric-level.
>
> ---
>
> ### Conclusion
>
> We appreciate the reviewer’s thoughtful comments. In summary, we:
>
> - Clarify that we follow the MCAR/MAR/MNAR generators of Zhao et al. (TDM) [1] and DiffPuter [2] and aim for robustness under these mechanisms, rather than explicit modeling of $p(M\mid X)$.
> - Reorganized the empirical presentation to use clear tables and move dense bar plots (with full explanations) to the appendix.
> - Added uncertainty measures and significance tests to quantify stability.
> - Provide new downstream experiments showing that Impute-MACFM’s imputation improvements translate into better predictive performance.
>
> We hope these clarifications and new results address your concerns and justify a more positive assessment of the paper.
>
> ---
>
> ### References
>
> [1] He Zhao, Ke Sun, Amir Dezfouli, and Edwin V. Bonilla. **Transformed distribution matching for missing value imputation.** In *International Conference on Machine Learning (ICML)*, pages 42159–42186. PMLR, 2023.
>
> [2] Zhang Hengrui, Fang Liancheng, Wu Qitian, and Philip S. Yu. **Empowering diffusion models for missing data imputation.** In *International Conference on Learning Representations (ICLR)*, 2025.

---

> > ### Comment · Reviewer_sxny · 2025-11-25
> >
> > I appreciate the responses and additional experiments from the authors. My concerns regarding uncertainty measures and downstream utility are addressed. However, my concerns for handling MNAR and MAR cases still remain. The authors explain how the MNAR- and MAR-masks can be generated for evaluation in the experiments section. However, the modeling itself only relies on the missing mask (which is also used by many existing works), and it remains unexplained why the proposed model could allow better imputation for MNAR and MAR. Given that better imputation for MNAR and MAR is argued to be a core contribution of the work, I will maintain my rating unchanged.

---

> ### Author Response · Authors · 2025-11-20
> **Downstream on classification datasets**
>
> To directly assess downstream utility, we train a **logistic regression** classifier on the imputed training data produced by each method and evaluate performance on the imputed test data. The following two tables report **classification accuracy** and **macro AUROC**, respectively, averaged over 5 runs (mean $\pm$ standard deviation) on five UCI datasets.
>
> **Downstream classification accuracy (logistic regression, mean $\pm$ std).**
>
> | Method         | Adult                      | Default                    | Magic                      | Shoppers                   | Bean                            |
> |----------------|----------------------------|----------------------------|----------------------------|----------------------------|---------------------------------|
> | GAIN           | $0.8137 \pm 0.003$         | $0.7937 \pm 0.002$         | $0.7660 \pm 0.004$         | $0.8746 \pm 0.002$         | $0.8606 \pm 0.005$              |
> | HyperImpute    | $0.8273 \pm 0.002$         | $0.7960 \pm 0.002$         | $0.7813 \pm 0.003$         | $0.8656 \pm 0.003$         | $0.8783 \pm 0.004$              |
> | ICE            | $0.8162 \pm 0.003$         | $0.7990 \pm 0.002$         | $0.7638 \pm 0.003$         | $0.8662 \pm 0.003$         | $\mathbf{0.9211 \pm 0.002}$     |
> | MICE           | $0.8051 \pm 0.004$         | $0.7964 \pm 0.003$         | $0.7471 \pm 0.005$         | $0.8610 \pm 0.003$         | $0.8910 \pm 0.004$              |
> | MIRACLE        | $0.8205 \pm 0.003$         | $0.7971 \pm 0.002$         | $0.7778 \pm 0.003$         | $0.8662 \pm 0.002$         | $0.9167 \pm 0.003$              |
> | MIWAE          | $0.8177 \pm 0.003$         | $0.7931 \pm 0.002$         | $0.7555 \pm 0.004$         | $0.8735 \pm 0.002$         | $0.7967 \pm 0.006$              |
> | Most-frequent  | $0.8174 \pm 0.001$         | $0.7952 \pm 0.002$         | $0.7564 \pm 0.001$         | $0.8754 \pm 0.002$         | $0.7933 \pm 0.001$              |
> | MissForest     | $0.8226 \pm 0.002$         | $0.7969 \pm 0.002$         | $0.7585 \pm 0.003$         | $0.8692 \pm 0.002$         | $0.8957 \pm 0.003$              |
> | SoftImpute     | $0.8138 \pm 0.002$         | $0.7950 \pm 0.002$         | $0.7659 \pm 0.003$         | $0.8708 \pm 0.002$         | $0.8966 \pm 0.003$              |
> | MCFlow         | $0.8023 \pm 0.003$         | $0.7954 \pm 0.003$         | $0.7326 \pm 0.005$         | $0.8440 \pm 0.004$         | $0.8540 \pm 0.004$              |
> | Remasker       | $0.8146 \pm 0.003$         | $0.8012 \pm 0.002$         | $0.7592 \pm 0.003$         | $0.8724 \pm 0.003$         | $0.8902 \pm 0.003$              |
> | GRAPE          | $0.8265 \pm 0.003$         | $0.7964 \pm 0.002$         | $0.7636 \pm 0.003$         | $0.8746 \pm 0.003$         | $0.8901 \pm 0.003$              |
> | DiffPuter      | $0.8225 \pm 0.003$         | $0.7949 \pm 0.002$         | $0.7785 \pm 0.003$         | $0.8635 \pm 0.003$         | $0.9028 \pm 0.003$              |
> | Impute-MACFM   | $\mathbf{0.8392 \pm 0.001}$ | $\mathbf{0.8028 \pm 0.002}$ | $\mathbf{0.7964 \pm 0.003}$ | $\mathbf{0.8802 \pm 0.001}$ | $0.9045 \pm 0.003$              |

---

> ### Author Response · Authors · 2025-11-20
>
> **Downstream macro AUROC (logistic regression, mean $\pm$ std over 5 runs).**
>
> | Method         | Adult                      | Default                    | Magic                      | Shoppers                   | Bean                            |
> |----------------|----------------------------|----------------------------|----------------------------|----------------------------|---------------------------------|
> | GAIN           | $0.7968 \pm 0.004$         | $0.7155 \pm 0.003$         | $0.8002 \pm 0.003$         | $0.8247 \pm 0.003$         | $0.9859 \pm 0.002$              |
> | HyperImpute    | $0.8726 \pm 0.002$         | $0.7261 \pm 0.003$         | $0.8184 \pm 0.002$         | $0.8330 \pm 0.002$         | $0.9897 \pm 0.001$              |
> | ICE            | $0.8517 \pm 0.003$         | $0.7203 \pm 0.002$         | $0.7852 \pm 0.004$         | $0.8160 \pm 0.003$         | $\mathbf{0.9944 \pm 0.001}$     |
> | MICE           | $0.8297 \pm 0.003$         | $0.7120 \pm 0.004$         | $0.7679 \pm 0.004$         | $0.7930 \pm 0.003$         | $0.9848 \pm 0.002$              |
> | MIRACLE        | $0.8626 \pm 0.002$         | $0.7207 \pm 0.003$         | $0.8101 \pm 0.003$         | $0.8239 \pm 0.002$         | $0.9925 \pm 0.001$              |
> | MIWAE          | $0.8458 \pm 0.003$         | $0.7140 \pm 0.003$         | $0.7871 \pm 0.003$         | $0.8282 \pm 0.003$         | $0.9637 \pm 0.003$              |
> | Most-frequent  | $0.8424 \pm 0.003$         | $0.7188 \pm 0.002$         | $0.7375 \pm 0.001$         | $0.8272 \pm 0.002$         | $0.9622 \pm 0.001$              |
> | MissForest     | $0.8648 \pm 0.002$         | $0.7206 \pm 0.002$         | $0.7899 \pm 0.003$         | $0.8427 \pm 0.002$         | $0.9909 \pm 0.001$              |
> | SoftImpute     | $0.8325 \pm 0.003$         | $0.7144 \pm 0.003$         | $0.7865 \pm 0.003$         | $0.8339 \pm 0.002$         | $0.9925 \pm 0.002$              |
> | MCFlow         | $0.8156 \pm 0.004$         | $0.7019 \pm 0.004$         | $0.7867 \pm 0.004$         | $0.6907 \pm 0.005$         | $0.9691 \pm 0.003$              |
> | Remasker       | $0.8713 \pm 0.002$         | $0.7333 \pm 0.003$         | $0.7902 \pm 0.003$         | $0.8443 \pm 0.002$         | $0.9745 \pm 0.002$              |
> | GRAPE          | $0.8766 \pm 0.002$         | $0.7177 \pm 0.003$         | $0.8017 \pm 0.003$         | $0.8274 \pm 0.002$         | $0.9830 \pm 0.001$              |
> | DiffPuter      | $0.8552 \pm 0.003$         | $0.7208 \pm 0.003$         | $0.8151 \pm 0.002$         | $0.8319 \pm 0.002$         | $0.9921 \pm 0.001$              |
> | Impute-MACFM   | $\mathbf{0.8818 \pm 0.002}$ | $\mathbf{0.7480 \pm 0.002}$ | $\mathbf{0.8445 \pm 0.003}$ | $\mathbf{0.8867 \pm 0.001}$ | $0.9918 \pm 0.002$              |

---

### Meta-Review · Area_Chair_CgFg · 2026-01-06

**Summary:**

The paper proposes a mask-aware flow-matching method for tabular imputation. Reviewers think the problem is important and the method is technically sound. However, there is a consensus that the novelty is limited and reviewers concerned that claims about handling MAR/MNAR missingness are overstated. Although the rebuttal improved clarity and added downstream results and uncertainty analysis, it seemed that a few reviewers kept their original rating in terms of the contribution and novelty.

**Reviewer Concerns:**

**Addressed by the rebuttal**

- Variance and stability analysis

- Downstream predictive experiments

- Improved table/figure formatting and paper clarity

**Still outstanding**

- Limited novelty, contribution viewed as engineering refinement rather than conceptual advance

- MNAR/MAR

**Reviewer Scores:**

Two reviewers explicitly stated to remain their negative ratings. Although the other two reviewers didn't reply to the authors' rebuttal, their main concern is on the novelty, which seems to be hard to be addressed by the rebuttal. Thus, I would expect they wouldn't have changed their score significantly.

---

### Decision · Program_Chairs · 2026-01-26

Reject